# The Vaccination Concerns in COVID-19 Scale (VaCCS): Development and validation

Kyra Hamilton[1,2,3]*, Martin S. Hagger[1,3,4,5]*

1 School of Applied Psychology, Griffith University, Brisbane, Australia, 2 Menzies Health Institute Queensland, Griffith University, Gold Coast, Australia, 3 Health Sciences Research Institute, University of California, Merced, California, United States of America, 4 Psychological Sciences, University of California, Merced, California, United States of America, 5 Faculty of Sport and Health Sciences, University of Jyväskylä, Jyväskylä, Finland

* mhagger@ucmerced.edu (MSH); khamilton4@ucmerced.edu (KH)

**Data Availability Statement:** Data files and analysis output related to the research are available online: https://osf.io/k96bn/.

**Funding:** The author(s) received no specific funding for this work.

## Abstract

Vaccines are highly effective in minimizing serious cases of COVID-19 and pivotal to managing the COVID-19 pandemic. Despite widespread availability, vaccination rates fall short of levels required to bring about widespread immunity, with low rates attributed to vaccine hesitancy. It is therefore important to identify the beliefs and concerns associated with vaccine intentions and uptake. The present study aimed to develop and validate, using the AMEE Guide, the Vaccination Concerns in COVID-19 Scale (VaCCS), a comprehensive measure of beliefs and concerns with respect to COVID-19 vaccines. In the scale development phase, samples of Australian ($N = 53$) and USA ($N = 48$) residents completed an initial open-response survey to elicit beliefs and concerns about COVID-19 vaccines. A concurrent rapid literature review was conducted to identify content from existing scales on vaccination beliefs. An initial pool of items was developed informed by the survey responses and rapid review. The readability and face validity of the item pool was assessed by behavioral science experts ($N = 5$) and non-experts ($N = 10$). In the scale validation phase, samples of Australian ($N = 522$) and USA ($N = 499$) residents completed scaled versions of the final item pool and measures of socio-political, health beliefs and outcomes, and trait measures. Exploratory factor analysis yielded a scale comprising 35 items with 8 subscales, and subsequent confirmatory factor analyses indicated acceptable fit of the scale structure with the data in each sample and factorial invariance across samples. Concurrent and predictive validity tests indicated a theoretically and conceptually predictable pattern of relations between the VaCCS subscales with the socio-political, health beliefs and outcomes, and trait measures, and key subscales predicted intentions to receive the COVID-19 vaccine. The VaCCS provides a novel measure to assess beliefs and concerns toward COVID-19 vaccination that researchers and practitioners can use in its entirety or select specific subscales to use according to their needs.

**Competing interests:** The authors have declared that no competing interests exist.

## Introduction

COVID-19 infections have had substantive global social, economic, and health impacts, and contributed considerably to excess deaths worldwide [1–3]. In response, governments have imposed restrictions on movement and gatherings, and introduced other preventive measures (e.g., mandatory face mask wearing, imposing physical distancing rules, encouraging hand hygiene practices) to minimize infection transmission [4, 5]. The rapid development of highly efficacious COVID-19 vaccines and mass mobilization to distribute them has assisted in reducing infection rates, cases needing hospitalization, and bringing the pandemic under control [6, 7]. However, success of COVID-19 vaccination programs is highly dependent on high uptake rates among the population to build widescale immunity. Modeling data suggests that rates of vaccination of above 80% are required for exponential reductions in infection rates and, particularly, numbers of serious cases, hospitalizations, and deaths due to COVID-19 [8, 9]. Based on these data, some governments have begun to ease coronavirus restrictions and reduce requirements to engage in preventive measures contingent on high rates of vaccination uptake [10–12].

While there has been considerable uptake of the COVID-19 vaccine in countries where it has been offered to all adults such as Australia, Singapore, UK, and many European Union countries, the world population rate of fully vaccinated individuals is less than 40%. In addition, countries such as USA, Russia, and South Africa are witnessing a slowing of vaccine uptake [13], and many programs are running out of individuals willing to be vaccinated as a consequence [14, 15]. One contributing factor may be vaccine hesitancy, which represents a psychological state of indecision with respect to getting vaccinated [16]. Research and media reports have indicated considerable hesitancy with respect to COVID-19 vaccines in many populations and it has been identified as a salient contributor to reduced rates of vaccine uptake [17–20]. The issue is compounded by increased attention being given to 'antivax' groups and conspiracy theorists in the popular media who propagate misinformation and misperceptions about the efficacy, safety, and side effects of the COVID-19 vaccines [21–23]. Vaccine hesitancy in general, and specific concerns and false beliefs with respect to the COVID-19 vaccines, have considerable potential to stymie vaccination program effectiveness.

In addition to vaccine hesitancy in general, other beliefs (e.g., social and moral norms) [24, 25] and concerns (e.g., trust in government) [26] specific to COVID-19 may also contribute to intentions to receive, and actual uptake of, COVID-19 vaccines. Furthermore, vaccine hesitancy itself is likely to have multiple belief-based determinants [27]. This has compelled research examining these beliefs and the extent to which they account for unique variance in COVID-19 vaccination intentions. Knowledge of such beliefs, and their links to vaccination intentions, is valuable because it will inform the development of messaging included in interventions developed by organizations tasked with maximizing vaccine coverage of COVID-19 vaccine programs. This is especially important given the emergence of new strains of the SARS-CoV-2 virus that causes COVID-19, and the potential of the need for people to receive vaccine booster shots in the face of waning immunity, both of which will necessitate ongoing vaccination programs and accompanying advocacy [28–30].

One limitation of research examining relations between vaccine hesitancy, other vaccine-related beliefs and concerns, and vaccine intentions in the context of the COVID-19 pandemic is the relative dearth of measures with good psychometric properties and adequate construct and concurrent validity. While some measures of vaccine hesitancy and intentions, and other vaccine-related beliefs more broadly, have been developed in the context of COVID-19 or adapted from other measures [29], there is, to date, no validated measure that captures a broad range of beliefs and concerns that would be expected to be associated with COVID-19

vaccination intentions and uptake. Current measures tend to focus on one or a narrow range of factors such as confidence, trust, or vaccine literacy, and neglect the fact that vaccine hesitancy is likely to be a function of multiple beliefs and intrapersonal factors [27, 31].

With respect to vaccine hesitancy, conceptual work and theory suggests that it has multiple determinants such as socio-political beliefs, particularly generalized distrust in authority and medical science [32, 33]; generalized traits which include personality and individual difference constructs that reflect generalized tendencies that impact behavior and decision-making [30, 34–36]; and health-related beliefs and side effect concerns, particularly in terms of the rapid development of COVID-19 vaccines and their authorization for emergency use [37, 38]. Effects of these determinants on COVID-19 vaccine hesitancy and vaccine intentions, are likely to be heightened or exacerbated by contextual factors including the rapid development of the vaccine and the high profile of vaccine scepticism in media outlets particularly those with populist, right-leaning political perspectives. Specifically, the expedited process of vaccine development, a process more rapid than any vaccine in history [39–41], is likely to contribute to concerns among the general public with respect to safety. Concerns over a perceived lack of rigor or extensiveness of trials to test long-term effectiveness, or even a perception that developers 'cut corners', may contribute to vaccine concerns. This is likely to be exacerbated by misinformation regarding vaccine development and as well as high profile publicised cases of side effects or associations with nosocomial conditions [42, 43].

Alongside this, the rise of a populist political agenda that is generally sceptical of science and perceives vaccination as governmental interference and overreach, is also likely to heighten concerns relating to COVID-19 vaccine safety and rigor in development [41, 44]. In addition, high-profile vaccine-sceptic personalities and influencers in right-wing media outlets and social media platforms also model vaccine hesitancy and convey an aura of credibility to misinformation on the COVID-19 vaccine to a wide spectrum of the general public [22, 45]. These contextual factors are likely to magnify concerns among individuals pre-disposed to be sceptical and to whom right-wing populist beliefs have the most appeal, particularly those with traits such as right-wing authoritarianism and social dominance orientation.

Taken together, these factors specific to COVID-19 make it a special case with respect to beliefs and concerns about the vaccine, potentially contributing to the slowing rates of vaccination uptake observed in this context. Given the multifactorial nature of concerns surrounding the COVID-19 vaccines, there is a need for a comprehensive measure, which also demonstrates good psychometric properties with sets of constructs representing these specific issues (i.e., socio-political beliefs, personality and individual difference constructs, and health-related beliefs and outcomes), that can inform future intervention design and evaluation for COVID-19 vaccination uptake.

## The present study

To date, there is no evidence-based measure that captures sets of beliefs and concerns that are expected to relate to COVID-19 vaccination intentions. This study addresses this evidence gap by developing and validating a comprehensive measure that captures these concerns, the Vaccination Concerns in COVID-19 Scale (VaCCS). The VaCCS will provide researchers and practitioners with a novel measure to assess beliefs and concerns toward COVID-19 vaccination and assist in identifying the determinants of vaccination intention and uptake, informing the development of messaging and interventions that may promote vaccination. Importantly, our approach to validation is intended to produce a scale that is flexible to use such that researchers and practitioners can use the scale in its entirety or select specific sub-scales to use according to their needs.

We followed a systematic, multi-step process to develop and validate the VaCCS [46], which was expected to yield a final scale comprising of sets of items representing multiple COVID-19 vaccine concerns and belief dimensions. We expect concurrent and predictive validity tests to indicate a conceptually predictable pattern of relations between the VaCCS subscales with the socio-political, trait measures, and health beliefs and outcomes. Specifically, we anticipate that the scale will capture beliefs and concerns including, but not limited to: concerns with respect to side effects and concerns over safety attributed to a perceived lack of testing and the rapid development of COVD-19 vaccines, conspiracy beliefs relating to this and other vaccines and about how COVID-19 emerged, generalized fear of vaccines, distrust in the government and particularly the pharmaceutical companies that have developed the vaccines, uncertainty over the outcomes of the COVID-19 pandemic, and general knowledge and understanding of how vaccines work. We therefore expect individuals with higher concerns in these areas to be more likely to hold political views and express general distrust in government, to have higher concerns about vaccines in general and concerns over medicines, lower risk perceptions with respect to COVID-19, and also greater trait-levels of neuroticism and lower conscientiousness.

## Method

### Participants and recruitment

We followed a systematic, multi-step design process using the AMEE Guide [46] to develop the VaCCS. The AMEE Guide presents a seven-step survey scale design process broadly consisting of a development phase (steps 1–6) and a validation phase (step 7) (see [46] for details). In the scale development phase, samples of Australian ($N$ = 53; $M_{Age}$ = 44.45, $SD_{Age}$ = 19.57, 36% female) and USA ($N$ = 48; $M_{Age}$ = 36.95, $SD_{Age}$ = 12.64, 58% female) residents were recruited via a research panel company to complete an online open response survey. To be eligible for inclusion, participants were required to be aged 18 years or older and have not received a COVID-19 vaccine. Participants received modest compensation for their participation based on expected completion time consistent with the panel company's published rates. In the final phase of scale development, to assess readability and face validity of scale items, a convenience sample of 10 laypersons ($M_{Age}$ = 39.00, $SD_{Age}$ = 18.86, 50% female) and 5 experts (4 females; comprising clinical and health psychologists, behavior change scientists, and a vaccination researcher) were recruited.

For the subsequent scale validation phase, samples of Australian ($N$ = 522, 60.7% female) and USA ($N$ = 499, 70.7% female) residents were recruited via an online research panel company. To be eligible for inclusion, participants were required to be aged 18 years or older and not having received a COVID-19 vaccine. Participants were not stratified by specific demographic variables as our primary goal was to recruit participants who had not yet been vaccinated so we opted to be more inclusive in the recruitment phase. Data were collected between May 14 and May 28, 2021. At the time of data collection, only Australian residents aged 50 years and over, Aboriginal and Torres Strait Islander peoples aged 18 years and older, those with underlying medical conditions, and those whose employment placed them at high risk of contracting or spreading COVID-19 were eligible to get the COVID-19 vaccine [47]. By contrast, USA residents aged 18 years and older were eligible. We also collected data from an additional sample of vaccinated USA residents ($N$ = 479, 56.8% female) between June 3 and June 7, 2021, which we used to replicate the validation procedures. Participants in this sample were required to have received both doses of an FDA-approved two-dose vaccine (i.e., Pfizer, Moderna) or the one-dose vaccine (i.e., Johnson and Johnson). Participants received modest compensation for their participation based on expected completion time consistent with the panel company's published rates. Sample characteristics are presented in Table 1.

**Table 1. Sample characteristics and descriptive statistics.**

| Variable | Sample | | | Difference tests |
|---|---|---|---|---|
| | **Australia** | **USA (unvaccinated)** | **USA (vaccinated)** | |
| Participants | 522 | 499 | 479 | |
| Age, *M* years (SD) | 47.40 (14.83) | 55.36 (14.36) | 52.14 (14.55) | $F(2,1497) = 1.313, p = .269$ |
| Gender, *n* (%) | | | | $\chi^2 (2) = 21.285, p < .001$[b] |
| Female | 317 (60.7) | 353 (70.7) | 272 (56.8) | |
| Male | 202 (38.7) | 144 (28.9) | 203 (42.4) | |
| Non-binary | 0 (0.0) | 1 (0.2) | 2 (0.4) | |
| Not specified/prefer not to answer | 3 (0.06) | 1 (0.2) | 2 (0.4) | |
| Employment status, *n* (%) | | | | $\chi^2 (8) = 98.585, p < .001$ |
| Currently unemployed/full-time caregiver | 122 (23.4) | 116 (23.3) | 78 (16.3) | |
| Part-time/casual employed | 121 (23.2) | 55 (11.0) | 44 (9.2) | |
| Currently employed full-time | 181 (34.7) | 149 (29.9) | 229 (47.8) | |
| Leave without pay/furloughed | 2 (0.04) | 2 (0.4) | 1 (0.2) | |
| Retired | 96 (18.4) | 177 (35.6) | 127 (26.5) | |
| Race, *n* (%) | | | | $\chi^2 (10) = 105.950, p < .001$ |
| Black | 3 (0.6) | 36 (7.2) | 19 (4.0) | |
| Caucasian/White | 411 (78.7) | 433 (86.8) | 421 (87.9) | |
| Asian (South-East Asia/South Asia) | 75 (14.4) | 13 (2.6) | 22 (4.6) | |
| Middle-Eastern | 8 (1.5) | 2 (0.4) | 1 (0.2) | |
| Other | 12 (2.3) | 13 (2.6) | 13 (2.7) | |
| Prefer not to answer | 13 (2.5) | 2 (0.4) | 3 (0.6) | |
| Income, *n* (%)[a] | | | | $\chi^2 (2) = 10.353, p = .006$[b] |
| Low income ($\leq$ US\$30,000/AU\$40,000) | 61 (11.7) | 59 (11.8) | 78 (16.4) | |
| High income (> US\$30,000/AU\$40,000) | 150 (28.7) | 135 (27.1) | 315 (66.0) | |
| Prefer not to answer | 311 (59.6) | 305 (61.1) | 84 (17.6) | |
| Education level, *n* (%) | | | | $\chi^2 (8) = 105.11, p < .001$ |
| Completed junior/lower/primary school | 23 (4.4) | 13 (2.6) | 4 (0.8) | |
| Completed senior/high/secondary school | 158 (30.3) | 192 (38.5) | 122 (25.5) | |
| Post-school vocational qualification/diploma | 144 (27.6) | 120 (24.1) | 66 (13.8) | |
| Undergraduate University degree | 140 (26.8) | 130 (26.1) | 171 (35.7) | |
| Postgraduate University degree | 57 (10.9) | 44 (8.8) | 116 (24.2) | |
| Previous diagnosis for COVID-19 | | | | $\chi^2 (2) = 35.221, p < .001$[b] |
| Yes | 6 (1.2) | 47 (9.4) | 38 (7.9) | |
| No | 516 (98.8) | 448 (89.8) | 440 (91.9) | |
| Prefer not to say | 0 (0.0) | 4 (0.8) | 1 (0.2) | |
| Current COVID-19 | | | | $\chi^2 (8) = 31.415, p < .001$ |
| Yes | 1 (0.2) | 1 (0.2) | 18 (3.8) | |
| No | 520 (99.6) | 496 (99.4) | 460 (96.0) | |
| Prefer not to say | 1 (0.2) | 2 (0.4) | 1 (0.2) | |

*Note.*

[a]Participants were given the choice of opting out of reporting their income.

[b]Analysis based on a binary dependent variable.

## Design and procedure

Approval for study procedures was granted prior to data collection from the Griffith University Human Research Ethics Committee (#2021/108). A systematic, multi-step design process using the AMEE Guide, which broadly consists of a development phase (steps 1–6) and a

validation phase (step 7), was adopted to develop an initial set of items for the VaCCS [46]. In the first few steps of the developmental phase (steps 1–4; [46]), participants from the initial Australia and the USA samples completed the open response survey comprising five structured questions [48] and reported their demographic information. This step was conducted to learn how the population of interest conceptualizes and describes a range of beliefs and concerns with respect to COVID-19 vaccines [46]. Responses were subjected to qualitative content analysis using NVivo 10 qualitative analysis software, which resulted in the extraction of 10 themes: behavior, protection, barriers, emotions, normality, safety, trust, development, efficacy, and social influences (see S1 File).

Next, and concurrent with the open response survey, a rapid review of the literature was conducted followed by a synthesis of the open response survey and literature review data to develop a comprehensive set of survey items for further evaluation. Specifically, these steps were conducted to ensure that the beliefs and concerns raised with respect to COVID-19 vaccines aligned with relevant prior research and theory, to identify existing survey scales or items that might be used or adapted, and to ensure survey items were written in accordance with current best practice and the language used was appropriate to the population of interest [46]. The search was conducted in February and March 2021 using OVID, Medline, Web of Science, and Embase, and also encompassed meta-analyses and systematic reviews investigating the measurement of vaccine beliefs (see S2 File for the search syntax). The search resulted in the identification of 448 records after removal of duplicates. After abstract and full text screening, 10 systematic reviews and meta-analyses were included, from which 63 measures of vaccine beliefs were extracted. We also included the Oxford Coronavirus Explanation, Attitudes, and Narratives survey (making it 64 measures), which was published subsequent to the included reviews [29]. A flowchart of the screening and inclusion process is presented in S3 File and a list of the extracted scales and their sources is presented in S4 File. A pool of 561 items was extracted from the 64 measures identified in the review process, with a total of 480 items after duplicate items were removed. Items were then subjected to a thematic analysis using NVivo 10 qualitative analysis software and subsequently sorted into one of 11 themes and entered into an excel spreadsheet. The themes were largely similar to the themes extracted from the open response survey: behavior, efficacy-protection, barriers, emotions, normality, safety, trust, development, importance, knowledge, and alternative medicine. The items were then examined by the lead investigators for relevance and appropriateness, resulting in the identification of 159 items considered eligible for inclusion in the VaCCS. These items were then further reviewed for similarity of content and expression and cross-referenced with the qualitative open response survey data. This process resulted in the retention of 63 items for further evaluation. Items were adjusted where necessary to make reference to COVID-19 vaccination (see Table 2).

In the final steps of the development phase (steps 5–6; [46]), to ensure the readability and face validity of the scale, a convenience sample of behavioral science experts ($N = 5$) and laypersons ($N = 10$) and were recruited in May 2021. After providing demographic information and informed consent, participants were asked to complete the 63 potential items, and then rate them for readability ("The items are easy to understand for the average person"), relevance ("The measure is relevant to assessing people's COVID-19 vaccine beliefs and concerns"), and suitability ("The measure is suitable for assessing individuals' COVID-19 vaccine beliefs and concerns") on a 5-point Likert type scale (1 = *strongly disagree* to 5 = *strongly agree*) developed by the authors. Participants were also given the opportunity to provide further comment in an open response format, which resulted in a small number of edits to item wording to improve clarity. Results indicated that both expert and lay-person groups found the scale easy to understand ($M = 4.80$, $SD = 0.42$; $M = 4.25$, $SD = 0.50$, respectively), relevant ($M = 5.00$, $SD = 0.00$;

**Table 2. Results of exploratory factor analysis of candidate pool of items of the Vaccination Concerns in COVID-19 Scale (VaCCS).**

| # | Item | Factor | | | | | | | | |
|---|------|---|---|---|---|---|---|---|---|---|
| | | 1 | 8 | 2 | 3 | 5 | 7 | 9 | 4 | 6 |
| 29 | **If I get the COVID-19 vaccine it will help to protect my family and friends against the coronavirus** | **.740** | | | | | | | | |
| 27 | **The COVID-19 vaccine will protect me from the coronavirus** | **.734** | | | | | | | | |
| 30 | The COVID-19 vaccine will strengthen the immune system against the coronavirus | .733 | | | | | | | | |
| 26 | **The COVID-19 vaccine will stop the spread of the coronavirus** | **.720** | | | | | | | | |
| 25 | **The COVID-19 vaccine is effective** | **.711** | | | | | | | | |
| 31 | If individuals like me get the COVID-19 vaccine it will save a large number of lives | .678 | | | | | | | | |
| 32 | The COVID-19 vaccine is likely to work for almost everyone | .676 | | | | | | | | |
| 28 | **The COVID-19 vaccine will reduce the severity of symptoms if I get the coronavirus** | **.612** | | | | | | | | |
| 33 | The COVID-19 vaccine is likely to work for me | .573 | | | | | | | | |
| 53 | **Getting the COVID-19 vaccine will help to get things back to normal** | **.542** | | | | | | | | |
| 54 | Getting the COVID-19 vaccine will help to ensure people can freely travel again | .533 | | | | | | | | |
| 55 | Getting the COVID-19 vaccine will help to ensure people can freely go out again | .529 | | | | | | | | |
| 42 | **It is important to get the COVID-19 vaccine so that outbreaks do not occur** | **.492** | | | | | | | | |
| 43 | **Getting the COVID-19 vaccine is important for the health of others in my community** | **.483** | | | | | | | | |
| 41 | It is important to get the COVID-19 vaccine to prevent coronavirus spreading in the community | .478 | | | | | | | | .304 |
| 52 | Getting the COVID-19 vaccine will give me complete freedom to get on with life just as before | .472 | | | | | | | | |
| 44 | Getting the COVID-19 vaccine is important for my health | .426 | | | | | | | | .316 |
| 40 | Getting the COVID-19 vaccine is important | .413 | | | | | | | | |
| 46 | Getting the COVID-19 vaccine makes me feel relieved | .357 | | | | | | | | |
| 4 | It is safe for a person to get the COVID-19 vaccine | .315 | | | | | | | | |
| 18 | **I trust the Government to give me reliable information on the benefits and risks of the COVID-19 vaccine** | | .838 | | | | | | | |
| 19 | **I trust Healthcare Providers and Health Professionals to give me reliable information on the benefits and risks of the COVID-19 vaccine** | | .817 | | | | | | | |
| 12 | **I trust the Government's conclusions that the COVID-19 vaccine is safe** | | .816 | | | | | | | |
| 20 | **I trust Scientists to give me reliable information on the benefits and risks of the COVID-19 vaccine** | | .809 | | | | | | | |
| 14 | **I trust Scientists' conclusions that the COVID-19 vaccine is safe** | | .798 | | | | | | | |
| 13 | **I trust Healthcare Providers' and Health Professionals' conclusions that the COVID-19 vaccine is safe** | | .788 | | | | | | | |
| 24 | **I trust vaccine manufacturers to give me reliable information on the benefits and risks of the COVID-19 vaccine** | | .607 | | | | | | | |
| 11 | The COVID-19 vaccine was proven safe before it was approved for use | | .519 | | | | | | | |
| 3 | **I am concerned about the side effects of the COVID-19 vaccine** | | | .882 | | | | | | |
| 2 | I am worried about the side effects of the COVID-19 vaccine | | | .867 | | | | | | |
| 1 | **I fear that the COVID-19 vaccine will cause side effects** | | | .853 | | | | | | |
| 6 | I am concerned about the safety of the COVID-19 vaccine | | | .833 | | | | | | |
| 5 | **I am worried about the safety of the COVID-19 vaccine** | | | .801 | | | | | | |
| 7 | The COVID-19 vaccine is risky | | | .573 | | | | | | |
| 10 | **The COVID-19 vaccine can cause the coronavirus in some people** | | | | .939 | | | | | |
| 9 | **The COVID-19 vaccine can give you a serious case of the very same virus you're trying to avoid** | | | | .922 | | | | | |
| 8 | **I can get the coronavirus from the COVID-19 vaccine** | | | | .855 | | | | | |
| 16 | **The COVID-19 vaccine safety data is often made up** | | | | | .613 | | | | |
| 17 | **People have been deceived about the safety of the COVID-19 vaccine** | | | | | .606 | | | | |
| 22 | **The COVID-19 vaccine is promoted mainly because of manufacturers' profit** | | | | | .568 | | | | |
| 23 | **The main reason for promoting the COVID-19 vaccine is for drug companies to make money** | | | | | .527 | | | | |
| 15 | The COVID-19 vaccine safety data is untrustworthy | | | | | .484 | | | | |
| 49 | **I am opposed to the COVID-19 vaccine because it goes against freedom of choice** | | | | | .475 | | | | |

*(Continued)*

**Table 2.** (Continued)

| # | Item | Factor | | | | | | | | |
|---|---|---|---|---|---|---|---|---|---|---|
| | | 1 | 8 | 2 | 3 | 5 | 7 | 9 | 4 | 6 |
| 21 | A lot of important information about the COVID-19 vaccine is not shared with the public | | | | | .471 | | | | |
| **48** | **I am afraid of getting the COVID-19 vaccine** | | | | | | .900 | | | |
| **47** | **I am fearful about getting the COVID-19 vaccine** | | | | | | .868 | | | |
| **45** | **Getting the COVID-19 vaccine makes me feel anxious** | | | | | | .586 | | | |
| 61 | Overall, I am hesitant about getting the COVID-19 vaccine | | | | | | .306 | | | |
| 37 | I am more likely to trust the COVID-19 vaccine once it has been around for a while | | | | | | | .561 | | |
| **35** | **The COVID-19 vaccine is too new so I should wait before deciding to get it** | | | | | | | .553 | | |
| **34** | **More time is needed to be able to fully investigate the true effects of the COVID-19 vaccine** | | | | | | | .482 | | |
| **39** | **I am afraid that the COVID-19 vaccine has not been successfully tested on enough people** | | | | | | | .474 | | |
| 38 | I am concerned that the COVID-19 vaccine has not been tested adequately | | | .311 | | | | .467 | | |
| 63 | I am uncertain whether or not I will get the COVID-19 vaccine when it is offered to me | | | | | | | .339 | | |
| 36 | The speed of developing and testing the COVID-19 vaccine means it will be unsafe | | | | | | | .302 | | |
| **56** | **I have access to all the information I need to make good decisions about getting the COVID-19 vaccine** | | | | | | | | .830 | |
| **57** | **Information about the COVID-19 vaccine is easy to understand** | | | | | | | | .699 | |
| **58** | **I don't have enough information about the COVID-19 vaccine to decide** | | | | | | | | .593 | |
| 51 | Getting the COVID-19 vaccine should be on a strictly voluntary basis | | | | | | | | | -.444 |
| 50 | Individual rights are more important than requirements to get the COVID-19 vaccine | | | | | .373 | | | | -.433 |
| 60 | When the COVID-19 vaccine is offered to me, I will get it straight away | | .394 | | | | | | | .428 |
| 59 | I will get the COVID-19 vaccine when it is offered to me | | | | | | | | | -.425 |
| 62 | When the COVID-19 vaccine is available to me I will refuse to get it | | .371 | | | | | | | .425 |
| | Proportion of Variance Explained | .112 | .088 | .075 | .045 | .044 | .038 | .030 | .030 | .030 |
| | Cumulative Variance Explained | .112 | .200 | .275 | .320 | .363 | .401 | .431 | .461 | .491 |

*Note.* Factor 1 = Efficacy; Factor 2 = Worry; Factor 3 = Cause; Factor 4 = Literacy; Factor 5 = Scepticism; Factor 7 = Fear; Factor 8 = Trust; Factor 9 = Uncertainty. Coefficients are standardized structure factor loadings after oblimin rotation. Loadings are presented in order of size and factors presented in order of variance explained. Loadings < .300 are suppressed for clarity. Items in bold font were selected for the final 35-item VaCCS scale.

$M = 4.75$, $SD = 0.50$, respectively), and suitable ($M = 4.20$, $SD = 1.14$; $M = 4.75$, $SD = 0.50$, respectively).

In the scale validation phase (step 7; [46]), participants completed an anonymized online survey comprising study measures alongside measures of socio-demographic variables and characteristics (age, gender, employment status, education level, race, income). This last step in the VaCCS design process was to check for adequate item variance, reliability, and convergent and discriminant validity with respect to other measures [46]. Study measures comprised the candidate VaCCS items identified in the development phase alongside measures of constructs and variables used to test the concurrent and predictive validity of the VaCCS. Specifically, participants completed sets of measures tapping socio-political, personality and individual difference constructs, and health beliefs and outcomes. Where a 'target' behavior was mentioned in the study measures (e.g., intention), items in the unvaccinated Australian and USA samples made reference to getting the COVID-19 vaccine, while measures in the USA vaccinated sample referred to getting a booster vaccine in future to control for emerging variants of the virus.

## Measures

In the scale validation phase, we assessed concurrent and predictive validity of the VaCCS by including measures of constructs and variables expected to exhibit a characteristic pattern of

relations with the VaCCS subscales. The concurrent measures were in three broad categories: socio-political beliefs, personality and individual difference constructs, and health beliefs and outcomes. Unless otherwise stated, final composite scale scores for each participant were computed by averaging scores on each item.

**VaCCS.** Participants completed scaled versions of the 63 candidate items for the VaCCS identified in the scale development phase. Participants provided their responses on 7-point scales (1 = *completely disagree* to 7 = *completely agree*). The full list of candidate items is presented in Table 2.

**Socio-political beliefs.** *Vaccine hesitancy*. Two measures of vaccine hesitancy were adopted in the present study. The Oxford COVID-19 vaccine hesitancy scale [29] was adopted to measure vaccine hesitancy on seven items (e.g., "When the COVID-19 vaccine is available to me...") with responses provided on seven point scales (e.g., 1 = *I will refuse to get it* to 7 = *I will get it straight away*). Lower scores were associated with higher vaccine hesitancy. A single item measure of vaccine hesitancy was also included ("Overall, how hesitant are you about getting the COVID-19 vaccine?"), with responses provided on a 7-point scale (1 = *not at all* to 7 = *very much*). Higher scores indicated a higher level of vaccine hesitancy.

*Trust in government*. Participants' trust in governmental organizations' handling of the COVID-19 vaccine program was measured using a scale developed by Grimmelikhuijsen and Knies [49]. The scale consists of nine items (e.g., "With regard to the COVID-19 vaccination program, do you agree that the Government in office in your country acts in the best interest of its citizens?") with responses provided on 7-point scales (1 = *totally disagree* to 7 = *totally agree*). Higher scores on the scale represented greater trust.

*Political orientation*. Participants' political orientation was measured using two items [50]. The first prompted participants to report where their political ideology sits on a scale ranging from "far left" to "far right". Participants were provided with the following guide: "In political matters, people talk of "the left" and "the right". Far left can be characterized by an emphasis on equality, progress, freedom, and reform with the economy driven by a cooperative collective agency. The far right can be characterized by an emphasis on order, tradition, nationalism, and authority with the economy driven by market forces. A centrist view would be considered a balance between social equality and social hierarchy with the economy balanced between regulation and free market." Responses were provided on a sliding scale ranging from 0 to 100, with lower scores representing "far left" views and higher scores representing "far right" views. A second item prompted participants to respond to the following item: "How would you describe your political orientation?" Responses were provided on a sliding scale ranging from 0 to 100, with lower scores representing "strongly progressive" and higher scores representing "strongly conservative" views.

*Free will beliefs*. Free will beliefs were measured using the free will subscale of the Free Will Inventory [51]. The scale comprises five items (e.g., "People always have free will") with responses provided on 5-point scales (1 = *strongly disagree* to 5 = *strongly agree*). Higher responses reflected greater free will beliefs.

*Populist attitudes*. Participants reported their populist views on six items (e.g., "Politicians in the Government in office in your country need to follow the will of the people") [52] with responses provided on 7-point scales (1 = *strongly disagree* to 7 = *strongly agree*). Higher scores represented greater trust.

*Conspiracy beliefs*. Three sets of conspiracy beliefs were measured. Participants' beliefs in conspiracy theories specific to the COVID-19 vaccine were measured using the COVID-19 conspiracy beliefs scale [53]. General beliefs in vaccine conspiracy theories were measured on the vaccine conspiracy beliefs scale [54]. The scales comprise six (e.g., "COVID-19 is intentionally presented as dangerous in order to mislead the public") and seven items (e.g., "Vaccine

safety data is often fabricated"), respectively with responses provided on 7-point scales (1 = *strongly disagree* to 7 = *strongly agree*). General conspiracy beliefs were assessed using the Conspiracy Mentality Questionnaire (CMQ) designed to assess individuals' tendency to engage in general conspiracy ideation [55]. The scale comprises five items (e.g., "I think that many very important things happen in the world, which the public is never informed about") with responses provided on 11-point scales (1 = *certainly not* to 11 = *certainly*). Higher scores on each scale represent greater beliefs in conspiracy theories.

*Vaccine denial*. Participants were prompted to report whether they had previously declined a prescribed vaccination ("Have you ever refused or elected to forego a doctor-recommended vaccine?"), with responses provided on a binary scale (1 = *no* and 2 = *yes*).

**Personality and individual difference constructs.**   *Personality*. Personality dimensions were measured using the Ten-Item Personality Inventory (TIPI) [56]. Participants were instructed to read a set of ten bi-polar adjectives representing the extroversion (e.g., "extraverted, enthusiastic" and "reserved, quiet"), agreeableness ("critical, quarrelsome" and "sympathetic, warm"), conscientiousness ("dependable, self-disciplined" and "disorganized, careless"), neuroticism ("anxious, easily upset" and "calm, emotionally stable"), and openness to experience ("open to new experiences, complex" and "conventional, uncreative") traits from the five-factor model of personality. They were then prompted to rate the extent to which each pair described them using 7-point scales (1 = *strongly disagree* to 7 = *strongly agree*). Higher scale scores represent greater endorsement of the trait.

*Intolerance of uncertainty*. Participants' general trait-level uncertainty was measured using the Intolerance of Uncertainty Scale—Short Form (IUS-12) [57]. The scale comprises 12 items (e.g., "Unforeseen events upset me greatly") and responses were provided on 6-point scales (1 = *not at all characteristic of me* to 6 = *entirely characteristic of me*). Higher scales scores reflect greater uncertainty.

**Health beliefs and outcomes.**   *Social cognition constructs*. Multi-item measures of intentions to receive the COVID-19 vaccine (e.g., "I intend to get the COVID-19 vaccine when it is offered to me?") and risk perceptions (e.g., "It would be risky for me to get the COVID-19 vaccine when it is offered to me?") with respect to getting the COVID-19 vaccine, were developed according to published guidelines [58]. Each measure made reference to getting the COVID-19 vaccine as the target behavior for the Australia and USA unvaccinated samples, and getting the COVID-19 'booster' vaccine in the vaccinated USA sample. Responses were provided on 7-point scales (1 = *strongly disagree* to 7 = *strongly agree*). Higher scores represented greater intentions and risk perceptions.

*Beliefs about medicines*. Participants beliefs and concerns about medicines were measured on the beliefs about medicines questionnaire [59]. The questionnaire comprises eight items (e.g., "Doctors use too many medicines") and responses are provided on 5-point scales (1 = *strongly disagree* to 5 = *strongly agree*). Higher scores represent more concerns or negative beliefs about medicines.

*Vaccine confidence*. Confidence in vaccinations was assessed using the vaccination confidence subscale of the "5C" Psychological Antecedents of Vaccination Scale [60]. The scale consists of five items (e.g., "Vaccination is unnecessary because vaccine preventable diseases are not common anymore") with responses provided on 7-point scales (1 = *strongly disagree* to 7 = *strongly agree*). Higher scores represent lower confidence in vaccines.

*Vaccine knowledge*. Participants' vaccine knowledge was assessed using a single-item vaccination knowledge scale [61]. The scale comprises nine items (e.g., Vaccines are superfluous, as diseases can be treated (e.g., with antibiotics)?"), with participants providing responses on a binary scale (e.g., 1 = *yes* or 0 = *no*). Participants also had the option of checking a "I don't know" option. The final scale score was computed by summing participants' "yes" responses.

*Close contact with vulnerable people*. Participants reported the likelihood they would come into close contact with people they considered particularly vulnerable to COVID-19 in their home, in their social groups, and in their workplace on three items (e.g., "Do you have people living in your household who are considered vulnerable to COVID-19?"). Participants provided their responses on a binary scale (e.g., 1 = *yes* or 0 = *no*). An overall vulnerability index was produced by summing the "yes" responses.

*Self-rating of health*. Self-rated health was measured with a single item: "In general, how would you rate your health?" [62]. Responses were provided on a 5-point scale (1 = *very poor* to 5 = *very good*). Higher scores reflect better health.

*Influenza vaccine history*. Participants reported whether or not they had received the influenza vaccine in the current year ("Did you get the seasonal influenza vaccination in the prior year?"), with responses provided on a binary scale (1 = *no* and 2 = *yes*), and whether or not they regularly received the influenza vaccine ("When the seasonal influenza vaccine is available each year I..."), with responses provided on a 7-point scale (1 = *never get it* and 7 = *always get it*).

**Socio-demographic variables.** Participants self-reported their age in years, sex (female, male, non-binary, not specified/prefer not to say), employment status (currently unemployed/ full time caregiver, currently full-time employed, part-time employed, on leave without pay/ furloughed, retired), annual household income stratified by eleven income levels based on Australia and USA national averages, highest level of formal education (completed junior/ lower/primary school, completed senior/high/secondary school, post-school vocational qualification/diploma, undergraduate university degree, postgraduate university degree), and race (Black, Caucasian/White, Asian, Middle-eastern, other, prefer not to say). Dichotomous employment (unemployed vs. employed), highest education level (completed school education only vs. completed post-school education), and race (Caucasian/White vs. non-White) variables were computed for use in subsequent analyses.

## Data analysis

**Construct validity.** *Exploratory factor analysis and item selection*. In the first instance, we subjected data on the initial pool of 63 items for the VaCCS to an exploratory factor analysis with the goal of identifying subscales that represent concerns and social cognition beliefs with respect to COVID-19 vaccine. The matrix of correlations among the item pool in the Australia and USA samples was subjected to a principal components analysis. In the analysis, factors with eigenvalues greater than 1.00 were extracted and subjected to oblimin rotation with Kaiser normalization. The standardized structure matrix of item factor loadings for each extracted factor was used as a basis for item selection for the final scale. Items were selected based on the size of their loading on their respective factor and their representativeness in terms of capturing the essence of the factor without duplication. Extracted factors were labeled based on the collective content of the items comprising each extracted factor. The exploratory factor analysis was conducted using the psych [63] and GPArotation [64] packages in R.

*Confirmatory factor analysis and factorial invariance*. The resulting scale was subjected to confirmatory factor analysis in the Australia and USA samples separately, followed by a multigroup analysis to test for invariance of the proposed factor structure across the samples. In each of the single-sample confirmatory factor analytic models, items pertaining to each factor selected based on the exploratory factor analysis and item review in the previous step were set as indicators of its respective factor with each factor set to correlate with each other. Fit of the models with the data in each sample was assessed using multiple goodness-of-fit criteria: the comparative fit index (CFI) and the Tucker-Lewis index (TLI), the root mean square error of

approximation (RMSEA) and its 90% confidence intervals, and the standardized root mean square residual (SRMSR). For a well-fitting model, the CFI and TLI should approach or exceed .900 for a well-fitting model, although values approaching or exceeding .950 are preferable, the RMSEA should be close to .008 with narrow confidence intervals, and the SRMSR should be close to .006 [65, 66]. We also examined solution estimates for the model in each sample including the standardized factor loadings with their 95% confidence intervals and the R-squared value for each item on its hypothesized factor, and the average variance extracted by the set of items for each factor. Factor loadings and R-squared values should be close to .700 and .500, respectively, and average variance extracted should exceed .500 for good structural integrity of the proposed latent factors [67, 68].

Pending adequate fit of the proposed confirmatory factor analysis model in each sample, we tested the invariance of the proposed factor structure across the Australia and USA samples. We followed a recommended invariance routine beginning with a configural model, in which the tenability of the proposed model across the two samples is tested, followed by progressively restrictive models in which the factor loadings ("weak invariance") and item intercepts ("strong invariance") are constrained to be equal across the samples. As the goodness-of-fit chi-square is recognized to be over-sensitive as a means to evaluate changes in model fit in the invariance routine, we followed Cheung and Rensvold's [69] recommendations that invariance is supported if the incremental fit indexes (CFI, TLI) of the constrained models differ from the configural model by less than .01.

Assuming adequacy of the model in the initial samples, we subsequently aimed to replicate these findings by fitting the confirmatory factor analytic model to data on the VaCCS items from a third sample of vaccinated participants from the USA. Fit of the model was evaluated using the same criteria for overall model fit and solution estimates in the unvaccinated samples. All confirmatory factor analytic models were estimated using the lavaan [70] package in R using a maximum likelihood estimation method and full-information maximum likelihood (FIML) imputation of missing data.

**Scale scores and reliability coefficients.** Item scores for the VaCCS subscales produced in the factor analyses were averaged to provide scale scores. We similarly computed scale scores for the sets of socio-political, health beliefs and outcomes, and personality and individual difference constructs. The only exceptions were the vulnerable people and COVID-19 knowledge measures; both measures adopted binary scales, so overall scale scores were computed using the sum rather than the average of the item scores. Omega ($\omega$) reliability coefficients were computed to evaluate the internal consistency of each scale with estimates equal to or in excess of .800 considered indicative of good internal consistency [71]. Where scales comprised fewer than three items, we computed the inter-item correlation in lieu of a reliability coefficient.

**Concurrent and predictive validity.** We examined correlations between the VaCCS subscales and the sets of socio-political, health beliefs and outcomes, and personality and individual difference constructs measured concurrently with the VaCCS items. Specifically, we correlated each composite scale from the VaCCS with composite scales from the scales from each set and in each sample separately. To minimize the false positivity rate (Type I error rate) due to multiple tests, we applied a Holm-Bonferroni correction method to adjust statistical significance levels. We used linear multiple regression to examine the unique contribution of each VaCCS subscale in predicting intentions to get the COVID-19 vaccine in the Australia and USA unvaccinated samples, and the COVID-19 'booster' vaccine in the USA vaccinated sample. Specifically, we simultaneously regressed measures of intentions on the VaCCS subscales, risk perceptions, vaccine hesitancy, and socio-demographic variables (age, gender, education, employment status, and race) in each sample. In order to avoid including multiple

contrast codes for the categorical socio-demographic variables with multiple categories in the regression model, we dichotomized variables with more than one category. Specifically, for the education variable participants were divided into those with primary education as the highest level of education (0) and those with secondary education or higher (1). Employment was divided into employed (1) and unemployed (0). Ethnicity was divided into non-white (1) and white (2) participants. We did not include income because of the large amount of missing data. The correlation and regression analyses were conducted using the psych and lavaan packages, respectively, in R.

## Results

### Participants

Sample characteristics for each sample are presented in Table 1. Participants in the Australia and unvaccinated USA samples were predominantly white (78.7% and 86.8%, respectively), educated (65.3% and 59%, respectively, reported completing a post-school qualification), predominantly employed in at least part-time work (57.9% and 40.9%, respectively), and reported middle-to-high income (28.7% and 27.1%, respectively). Participants in the vaccinated USA sample were majority white (87.9%), employed in at least part-time work (56.0%), educated (73.7% reported completing a post-school qualification), and reported high-to-middle income (66.0%). Very few participants reported having a previous diagnosis for COVID-19 in the Australia sample (1.2%), and self-reported previous COVID-19 positivity rates were also low in the vaccinated (9.4%) and unvaccinated (7.9%) USA samples. Only one case (0.2%) in each of the Australia and unvaccinated USA samples reported a current COVID-19 diagnosis, compared to 3.8% in the vaccinated USA sample. The majority of the vaccinated USA sample reported receiving a two-dose vaccine regimen (Pfizer, 48.6%; Moderna, 36.3%) with the remainder receiving the Jansen/Johnson & Johnson vaccine (12.9%), or did not know the type of vaccine they received (1.1%) or preferred not to say (1.1%).

Examining differences in sample demographic characteristics across the three samples revealed no significant differences in age, $F(2,1497) = 1.313$, $p = .269$. However, we found statistically significant differences in the distribution of participants across categories of gender ($\chi^2 (2) = 21.285$, $p < .001$), employment status ($\chi^2 (8) = 98.585$, $p < .001$), race ($\chi^2 (10) = 105.950$, $p < .001$), income ($\chi^2 (2) = 10.353$, $p = .006$), education level ($\chi^2 (8) = 105.11$, $p < .001$), previous COVID-19 diagnosis ($\chi^2 (2) = 35.221$, $p < .001$), and current COVID-19 diagnosis ($\chi^2 (8) = 31.415$, $p < .001$). Follow-up tests revealed an equal distribution of gender across the USA unvaccinated and vaccinated samples, but the Australian sample had a larger proportion of men than the two USA samples. In addition, proportions of unemployed individuals, retired individuals, and individuals in part-time and full-time work differed across all samples. The demographic profile of the Australia and USA vaccinated sample did not differ, but these samples had significantly larger numbers of black participants and fewer Asian participants than the USA sample. There were no differences in the distribution of individuals in high and low income categories across the Australia and USA unvaccinated samples, but there were significantly more participants in the high income category in the USA vaccinated sample relative to these samples. While there were only marginal differences in proportions of participants across education categories between the Australia and USA unvaccinated samples, the USA vaccinated sample had greater proportions of participants in the higher education categories. Finally, the Australia and the USA vaccinated sample reported a greater number of past COVID-19 diagnoses relative to the USA unvaccinated sample, while the USA vaccinated sample reported a higher proportion of current COVID-19 diagnoses than the Australia and USA unvaccinated sample. These data indicate systematic differences in the demographic

characteristics of the three samples, and highlight the imperative of including demographic characteristics as covariates in subsequent cross-national comparisons.

## Exploratory factor analysis and item selection

Nine factors explaining 49.1% of the variance in correlations among the initial pool of 63 VaCCS items were extracted in the exploratory factor analysis. The rotated structure matrix with standardized factor loadings of the VaCCS items is presented in Table 2. The content of each factor is described next in the order of variance explained rather than position extracted. The first extracted factor explained 11.2% of the variance and comprised items relating to the efficacy of the COVID-19 vaccine to protect the self and others from infection (e.g., "If I get the COVID-19 vaccine it will help to protect my family and friends against the coronavirus") and to assist in returning society to normal (e.g., "Getting the COVID-19 vaccine will help to get things back to normal"). This factor was labelled "Beliefs in efficacy and prevention". The eighth extracted factor explained 8.8% of the variance and consisted of items reflecting trust in the government and other organizations with respect to the COVID-19 vaccine (e.g., "I trust the Government to give me reliable information on the benefits and risks of the COVID-19 vaccine"). This factor was labelled "Trust in authorities". The second extracted factor explained 7.5% of the variance and comprised items reflecting concerns about the safety and side effects of the vaccine (e.g., "I am concerned about the side effects of the COVID-19 vaccine"). This factor was labelled "Worry about safety and side effects". The third extracted factor accounted for 4.5% of the variance and consisted of items reflecting concerns that the vaccine may cause the virus in some people (e.g., "The COVID-19 vaccine can cause the coronavirus in some people"). The factor was labelled "Beliefs that the vaccine causes COVID-19". The fifth extracted factor explained 4.4% of the variance and comprised items reflecting scepticism and lack of trust in the vaccine (e.g., "The COVID-19 vaccine safety data is often made up"). This factor was labelled "Scepticism and mistrust in the vaccine". The seventh extracted factor explained 3.8% of the variance and consisted of items reflecting fear and anxiety over getting the vaccine (e.g., "I am afraid of getting the COVID-19 vaccine"). This factor was labelled "Fear of the vaccine". The ninth extracted factor accounted for 3.0% of the variance and comprised items reflecting uncertainty about the vaccine development and hesitancy in getting vaccinated until it had been further evaluated (e.g., "The COVID-19 vaccine is too new so I should wait before deciding to get it"). This factor was labelled "Uncertainty and hesitation in getting vaccinated". The fourth extracted factor also explained 3.0% of the variance and comprised items relating to adequacy and understanding of information with respect to the COVID-19 vaccine ("Information about the COVID-19 vaccine is easy to understand"). This factor was labelled "Vaccine literacy"). The sixth extracted factor comprised items relating to rights, uptake of the vaccine, and refusal to take the vaccine. Items pertaining to this factor exhibited much lower principal loadings and cross loadings of similar magnitude, so it was dropped in the final reckoning for the VaCCS.

A final item selection procedure for each included subscale was subsequently applied by each lead researcher independently based on three criteria: representativeness of the items, statistical evidence from the exploratory factor analysis (i.e., factor loadings), and the need for parsimony. The procedure resulted in close agreement across the researchers in item selection (average 87.79% agreement, AC1/AC2 agreement statistic = 0.781) [72]. Discrepancies were discussed and the selection criteria and procedure updated accordingly. Subsequent re-application of the procedure to the candidate items by each researcher resulted in perfect agreement for each subscale (100%). The procedure yielded a final scale comprising of 35 items and eight subscales.

## Confirmatory factor analyses and invariance analyses

Next, we fit the 8-factor, 35-item VaCCS model to data from the Australia and USA samples separately using confirmatory factor analysis. Pending adequate fit of the model with the data in each sample, we subsequently tested the invariance of the model factor loadings and intercepts across the two samples using multi-group confirmatory factor analysis. We also replicated the confirmatory factor analytic model in the additional USA sample of vaccinated participants.

Goodness-of-fit statistics for the initial confirmatory factor analysis model in each sample are presented in Table 3. Overall, the model exhibited fit statistics that approached acceptable cut-off values for the multiple adopted indices of good fit. However, modification indices (MI) indicated that redundancy across some items from within the same subscale was primarily responsible for the misspecification, and identified pairs of within-factor indicator error covariances that would resolve the misspecification if freely estimated. The error variances associated with the largest misspecification in the model were identical across the two samples. Examination of the content of these items indicated that they tapped similar conceptual content in the measured factor. We therefore set error covariances free between items with the five largest expected parameter change (EPC) statistics identified by the MIs (EPC > 50.000) in each sample. The practice of including error covariances is only advocated when justified by the content of the items involved in the redundancy [73]. In the case of the inclusion of the five error covariances for these models, there was justification given the closeness in the item content. We expected the inclusion of these error covariances would have little consequence on model integrity and we did not want reject a tenable model on trivial grounds. The final model including these error covariances exhibited acceptable goodness-of-fit statistics in both samples (Table 3). Solution estimates of the final model in each sample are presented in Table 4. Factor loadings and R-squared values for each item on its respective factor approached or exceeded the .70 and .50 expected values, respectively, with narrow confidence intervals for the factor loadings.

**Table 3. Model fit statistics for single-sample and multi-group confirmatory factor analyses with comparisons.**

| Samples and model | $\chi^2$ | df | CFI | TLI | RMSEA | 90% CI RMSEA | | SRMSR | $\Delta\chi^2$ | $\Delta$df | $\Delta$CFI | $\Delta$TLI |
|---|---|---|---|---|---|---|---|---|---|---|---|---|
| | | | | | | LB | UB | | | | | |
| Australia sample | | | | | | | | | | | | |
| Initial model | 2561.737*** | 532 | 0.904 | 0.892 | .087 | .084 | .091 | .048 | – | – | – | – |
| Final model | 1699.744*** | 526 | 0.947 | 0.940 | .065 | .062 | .069 | .043 | – | – | – | – |
| USA sample (unvaccinated) | | | | | | | | | | | | |
| Initial model | 2083.983*** | 532 | 0.915 | 0.905 | .076 | .073 | .080 | .053 | – | – | – | – |
| Final model | 1379.883*** | 526 | 0.953 | 0.947 | .057 | .053 | .061 | .046 | – | – | – | – |
| USA sample (vaccinated) | | | | | | | | | | | | |
| Initial model | 2687.983*** | 532 | 0.880 | 0.865 | .092 | .089 | .095 | .047 | – | – | – | – |
| Final model | 1599.621*** | 526 | 0.940 | 0.932 | .065 | .062 | .069 | .040 | – | – | – | – |
| Invariance analysis | | | | | | | | | | | | |
| Configural | 3211.064*** | 1056 | 0.946 | 0.940 | .063 | .061 | .066 | .044 | – | – | – | – |
| Weak invariance | 3275.088*** | 1083 | 0.946 | 0.940 | .063 | .061 | .065 | .047 | 64.025*** | 27 | .000 | .000 |
| Strong invariance | 3520.574*** | 1110 | 0.940 | 0.936 | .065 | .063 | .068 | .050 | 245.486*** | 27 | .006 | .004 |

*Note.* $\chi^2$ = Model goodness-of-fit chi-square; df = Degrees of freedom of model goodness of fit chi square; CFI = Comparative fit index; TLI = Tucker-Lewis Index; RMSEA = Root mean square error of approximation; 90% CI RMSEA; 90% confidence interval of the RMSEA; LB = Lower bound of the 90% confidence interval of the RMSEA; UB = Upper bound of the 90% confidence interval of the RMSEA; SRMSR = Standardized root mean square of the residuals; $\Delta\chi^2$ = Change in goodness-of-fit chi-square; $\Delta\chi^2$ = Change in degrees of freedom of the goodness of fit chi-square; $\Delta$CFI = Change in CFI; $\Delta$TLI = Change in TLI.

**Table 4. Results of confirmatory factor analysis of the Vaccination Concerns in COVID-19 Scale (VaCCS) in each sample.**

| Scale and Item# | Sample | | | | | | | | | | | | | | |
|---|---|---|---|---|---|---|---|---|---|---|---|---|---|---|---|
| | Australia | | | | | USA (unvaccinated) | | | | | USA (vaccinated) | | | | |
| | $\lambda$ | 95% CI | | $R^2$ | AVE | $\lambda$ | 95% CI | | $R^2$ | AVE | $\lambda$ | 95% CI | | $R^2$ | AVE |
| | | LB | UB | | | | LB | UB | | | | LB | UB | | |
| Efficacy | | | | | .735 | | | | | .739 | | | | | .593 |
| 29 | .886 | .865 | .906 | .784 | | .868 | .844 | .892 | .753 | | .839 | .807 | .870 | .703 | |
| 27 | .878 | .856 | .900 | .771 | | .912 | .895 | .930 | .833 | | .756 | .714 | .799 | .572 | |
| 26 | .846 | .820 | .873 | .716 | | .881 | .859 | .903 | .777 | | .762 | .720 | .804 | .580 | |
| 25 | .881 | .859 | .902 | .776 | | .882 | .860 | .903 | .777 | | .821 | .787 | .855 | .674 | |
| 28 | .770 | .733 | .807 | .593 | | .793 | .759 | .828 | .629 | | .766 | .725 | .807 | .586 | |
| 53 | .828 | .800 | .857 | .686 | | .824 | .794 | .855 | .680 | | .765 | .724 | .807 | .586 | |
| 42 | .876 | .853 | .898 | .767 | | .865 | .840 | .889 | .748 | | .726 | .679 | .773 | .527 | |
| 43 | .877 | .854 | .899 | .768 | | .850 | .824 | .877 | .723 | | .750 | .706 | .794 | .562 | |
| Trust | | | | | .819 | | | | | .793 | | | | | .705 |
| 18 | .892 | .872 | .911 | .795 | | .855 | .830 | .881 | .732 | | .781 | .744 | .818 | .610 | |
| 19 | .911 | .894 | .928 | .830 | | .841 | .814 | .868 | .707 | | .881 | .858 | .903 | .776 | |
| 12 | .944 | .932 | .955 | .891 | | .910 | .893 | .927 | .828 | | .861 | .836 | .887 | .742 | |
| 20 | .904 | .886 | .921 | .817 | | .897 | .878 | .916 | .805 | | .910 | .892 | .928 | .829 | |
| 14 | .928 | .915 | .942 | .862 | | .960 | .951 | .969 | .921 | | .931 | .916 | .946 | .866 | |
| 13 | .937 | .925 | .950 | .879 | | .939 | .927 | .951 | .882 | | .878 | .855 | .901 | .771 | |
| 24 | .792 | .758 | .825 | .627 | | .812 | .781 | .844 | .660 | | .668 | .617 | .720 | .446 | |
| Worry | | | | | .864 | | | | | .845 | | | | | .741 |
| 3 | .953 | .941 | .964 | .907 | | .957 | .944 | .970 | .916 | | .857 | .827 | .886 | .734 | |
| 1 | .940 | .927 | .953 | .883 | | .915 | .898 | .933 | .838 | | .783 | .744 | .823 | .614 | |
| 5 | .895 | .876 | .915 | .802 | | .887 | .865 | .909 | .786 | | .940 | .918 | .961 | .883 | |
| Cause | | | | | .813 | | | | | .800 | | | | | .781 |
| 10 | .950 | .935 | .966 | .903 | | .941 | .922 | .959 | .885 | | .898 | .877 | .920 | .807 | |
| 9 | .945 | .930 | .961 | .893 | | .915 | .894 | .936 | .837 | | .965 | .951 | .979 | .932 | |
| 8 | .809 | .776 | .841 | .654 | | .823 | .791 | .855 | .677 | | .796 | .760 | .831 | .633 | |
| Scepticism | | | | | .651 | | | | | .618 | | | | | .700 |
| 16 | .799 | .763 | .836 | .639 | | .753 | .708 | .798 | .567 | | .815 | .781 | .849 | .664 | |
| 17 | .823 | .789 | .857 | .677 | | .844 | .807 | .881 | .713 | | .840 | .810 | .870 | .706 | |
| 22 | .806 | .771 | .842 | .650 | | .814 | .777 | .852 | .663 | | .831 | .800 | .862 | .690 | |
| 23 | .811 | .777 | .846 | .658 | | .773 | .730 | .816 | .598 | | .846 | .817 | .876 | .716 | |
| 49 | .795 | .759 | .832 | .633 | | .753 | .709 | .796 | .566 | | .851 | .823 | .880 | .725 | |
| Fear | | | | | .841 | | | | | .769 | | | | | .832 |
| 48 | .962 | .951 | .973 | .925 | | .957 | .941 | .972 | .915 | | .949 | .936 | .962 | .900 | |
| 47 | .948 | .936 | .960 | .899 | | .960 | .945 | .975 | .922 | | .950 | .937 | .963 | .902 | |
| 45 | .830 | .802 | .859 | .690 | | .677 | .627 | .726 | .458 | | .832 | .803 | .862 | .693 | |
| Uncertainty | | | | | .766 | | | | | .676 | | | | | .659 |
| 35 | .894 | .872 | .917 | .800 | | .826 | .791 | .861 | .682 | | .832 | .800 | .864 | .692 | |
| 34 | .788 | .751 | .824 | .620 | | .849 | .817 | .882 | .721 | | .657 | .602 | .711 | .431 | |
| 39 | .918 | .898 | .938 | .842 | | .799 | .761 | .838 | .639 | | .924 | .903 | .944 | .853 | |
| Literacy | | | | | .602 | | | | | .541 | | | | | .736 |
| 56 | .854 | .819 | .888 | .729 | | .769 | .710 | .829 | .592 | | .829 | .796 | .863 | .688 | |
| 57 | .874 | .840 | .908 | .764 | | .794 | .733 | .856 | .631 | | .851 | .821 | .882 | .725 | |

(*Continued*)

**Table 4.** (Continued)

| Scale and Item# | Sample | | | | | | | | | | | | | | | | |
|---|---|---|---|---|---|---|---|---|---|---|---|---|---|---|---|---|---|
| | Australia | | | | | USA (unvaccinated) | | | | | USA (vaccinated) | | | | | | |
| | λ | 95% CI | | R² | AVE | λ | 95% CI | | R² | AVE | λ | 95% CI | | R² | AVE | | |
| | | LB | UB | | | | LB | UB | | | | LB | UB | | | | |
| 58 | .626 | .566 | .686 | .392 | | .654 | .586 | .722 | .428 | | .900 | .875 | .925 | .810 | | | |

*Note*. Efficacy = Beliefs in efficacy and prevention VaCCS subscale; Trust = Trust in authorities VaCCS subscale; Worry = Worry about safety and side effects VaCCS subscale; Cause = Beliefs vaccine causes COVID-19 VaCCS subscale; Scepticism = Scepticism and mistrust in the vaccine VaCCS subscale; Fear = Fear of vaccine VaCCS subscale; Uncertainty = Uncertainty and hesitation getting vaccinated VaCCS subscale; Literacy = Vaccine literacy VaCCS subscale; λ = Standardized factor loading; 95% CI = 95% Confidence interval of the standardized factor loading; LB = Lower bound of 95% confidence interval; UB = Upper bound of 95% confidence interval; AVE = Average variance extracted for the factor.

Given the acceptable fit of the single-sample models, we conducted the invariance analysis of the proposed model according to our a priori invariance routine. The configural model exhibited adequate goodness-of-fit with the data, indicating that the proposed model was tenable across the two samples (Table 3). Estimating models that constrained the factor loadings (weak invariance) and intercepts (strong invariance) revealed minimal differences in model fit across each model in the invariance routine according to Cheung and Rensvold's criterion for change in the CFI and TLI. We subsequently concluded that there was minimal variance in parameters of the proposed VaCCS model across the two samples.

Finally, we replicated the proposed confirmatory factor analytic model of the VaCCS in the USA vaccinated sample. Consistent with the previous analyses, the model exhibited acceptable goodness-of-fit statistics (Table 3) and solution estimates (Table 4) in this additional sample.

### Descriptive statistics and reliability estimates

Descriptive statistics and reliability estimates for the VaCCS subscales and study measures are presented in Table 5, and factor and manifest variable correlations among the subscales presented in Table 6. Reliability estimates indicated acceptable internal consistency for all VaCCS subscales. Reliability estimates for the variables included to test concurrent and criterion validity of the VaCCS were also acceptable, with few exceptions. The most notable exceptions were the two-item scales tapping the five-factor personality constructs, which all exhibited statistically significant inter-item correlations, but each was relatively modest in size, a trend noted elsewhere [56].

**Table 5. Descriptive statistics and reliability estimates of study measures.**

| Scale | # Items | Range | Australia sample | | | USA sample (unvaccinated) | | | USA sample (vaccinated) | | |
|---|---|---|---|---|---|---|---|---|---|---|---|
| | | | *M* | *SD* | ω | *M* | *SD* | ω | *M* | *SD* | ω |
| VaCCS Efficacy | 8 | 1−7 | 4.914 | 1.379 | .974 | 3.707 | 1.495 | .971 | 5.856 | 0.946 | .955 |
| VaCCS Trust | 7 | 1−7 | 4.554 | 1.604 | .981 | 3.095 | 1.620 | .977 | 5.551 | 1.207 | .969 |
| VaCCS Worry | 3 | 1−7 | 4.884 | 1.694 | .950 | 5.611 | 1.481 | .943 | 3.101 | 1.681 | .931 |
| VaCCS Cause | 3 | 1−7 | 2.943 | 1.572 | .950 | 3.744 | 1.643 | .943 | 2.429 | 1.583 | .931 |
| VaCCS Scepticism | 5 | 1−7 | 3.312 | 1.566 | .943 | 4.491 | 1.559 | .955 | 2.593 | 1.534 | .960 |
| VaCCS Fear | 3 | 1−7 | 4.270 | 1.785 | .939 | 5.037 | 1.682 | .905 | 2.743 | 1.657 | .938 |
| VaCCS Uncertainty | 3 | 1−7 | 4.775 | 1.657 | .904 | 5.557 | 1.406 | .866 | 3.379 | 1.541 | .857 |
| VaCCS Literacy | 3 | 1−7 | 4.466 | 1.426 | .833 | 3.906 | 1.589 | .792 | 5.621 | 1.204 | .897 |
| Oxford scale | 7 | 1−5 | 3.487 | 1.238 | .981 | 2.292 | 1.194 | .976 | 4.421 | 0.778 | .960 |

*(Continued)*

**Table 5.** (*Continued*)

| Scale | # Items | Range | Australia sample | | | USA sample (unvaccinated) | | | USA sample (vaccinated) | | |
|---|---|---|---|---|---|---|---|---|---|---|---|
| | | | *M* | *SD* | ω | *M* | *SD* | ω | *M* | *SD* | ω |
| Gov. trust | 9 | 1–7 | 4.522 | 1.636 | .989 | 3.042 | 1.778 | .991 | 5.125 | 1.494 | .986 |
| Polit. orient. | 2 | 0–100 | 49.330 | 19.694 | .652[a] | 63.685 | 23.866 | .704[a] | 49.289 | 25.383 | .806[a] |
| Free will | 5 | 1–7 | 5.179 | 1.094 | .907 | 5.190 | 1.260 | .921 | 5.339 | 1.080 | .915 |
| Polit. trust | 6 | 1–7 | 5.039 | 0.974 | .883 | 5.538 | 1.050 | .895 | 5.236 | 0.929 | .856 |
| COVID-19 CB | 6 | 1–7 | 3.005 | 1.369 | .916 | 4.242 | 1.471 | .906 | 2.788 | 1.292 | .916 |
| Vaccine CB | 7 | 1–7 | 3.229 | 1.521 | .974 | 4.452 | 1.544 | .969 | 2.736 | 1.551 | .978 |
| General CB | 5 | 1–11 | 6.829 | 2.066 | .894 | 8.011 | 1.888 | .888 | 6.612 | 2.002 | .895 |
| Vaccine hesitancy | 1 | 1–5 | 3.163 | 1.501 | – | 4.036 | 1.317 | – | 2.238 | 1.507 | – |
| Vaccine denial | | | | | | | | | | | |
| Intention | 3 | 1–7 | 4.568 | 1.973 | .991 | 2.674 | 1.878 | .991 | 5.995 | 1.209 | .976 |
| Risk perception | 2 | 1–7 | 4.063 | 1.716 | .767[a] | 5.039 | 1.581 | .801[a] | 2.753 | 1.588 | .768[a] |
| BMQ | 8 | 1–5 | 2.767 | 0.811 | .914 | 3.046 | 0.817 | .903 | 2.580 | 0.842 | .919 |
| Vaccine confidence | 5 | 1–5 | 2.707 | 0.653 | .609 | 2.856 | 0.604 | .456 | 2.243 | 0.696 | .766 |
| Vaccine knowledge[b] | 9 | 0–18 | 10.082 | 4.621 | .852 | 7.964 | 4.493 | .826 | 10.795 | 4.176 | .824 |
| Vulnerable people[b] | 3 | 1–6 | 5.100 | 1.128 | .834 | 5.018 | 1.268 | .842 | 4.835 | 1.078 | .790 |
| SRH | 1 | 1–5 | 3.634 | 0.955 | – | 3.593 | 1.057 | – | 3.818 | .959 | – |
| E | 2 | 1–7 | 3.333 | 1.312 | .429[a] | 3.645 | 1.503 | .414[a] | 3.722 | 1.475 | .383[a] |
| A | 2 | 1–7 | 5.066 | 1.080 | .230[a] | 5.252 | 1.120 | .183[a] | 5.168 | 1.158 | .199[a] |
| C | 2 | 1–7 | 5.383 | 1.136 | .457[a] | 5.743 | 1.086 | .511[a] | 5.676 | 1.141 | .419[a] |
| N | 2 | 1–7 | 3.467 | 1.385 | .542[a] | 3.127 | 1.360 | .451[a] | 3.204 | 1.417 | .518[a] |
| O | 2 | 1–7 | 4.455 | 1.174 | .291[a] | 4.670 | 1.230 | .317[a] | 4.694 | 1.201 | .216[a] |
| IUS-12 | 12 | 1–6 | 3.692 | 0.822 | .741[a] | 3.500 | 0.868 | .742[a] | 3.637 | 0.814 | .687[a] |

Note

[a]Reliability estimate for two-item scales is the Spearman rank-order inter-item correlation

[b]Scales comprises items with responses made on dichotomous (1 = *yes*, 0 = *no*) scales. ω = Revelle's Omega reliability coefficient; Efficacy = Beliefs in efficacy and prevention VaCCS subscale; Trust = Trust in authorities VaCCS subscale; Worry = Worry about safety and side effects VaCCS subscale; Cause = Beliefs vaccine causes COVID-19 VaCCS subscale; Scepticism = Scepticism and mistrust in the vaccine VaCCS subscale; Fear = Fear of vaccine VaCCS subscale; Uncertainty = Uncertainty and hesitation getting vaccinated VaCCS subscale; Literacy = Vaccine literacy VaCCS subscale; Oxford scale = Oxford COVID-19 Vaccine Hesitancy Scale (Freeman et al., 2021); Gov. trust = Trust in government organizations in handling COVID-19 (Grimmelikhuijsen & Knies, 2017); Polit. orient. = Political orientation (Kroh, 2007); Free will = Free will beliefs; Polit. trust = Trust in politicians handling of the COVID-19 pandemic; COVID-19 CB = COVID-19 conspiracy beliefs scale (Imhoff & Lamberty 2020); Vaccine CB = Vaccine conspiracy beliefs scale (Shapiro et al., 2016); General CB = General conspiracy beliefs scale (Bruder et al., 2013); Vaccine hesitancy = Single-item vaccine hesitancy measure; Vaccine denial = Single-item vaccine denial measure; Intention = COVID-19 vaccination intentions; Risk perception = Beliefs in risk of COVID-19; BMQ = Beliefs about medicines questionnaire; Vaccine confidence = Vaccine confidence scale (Betsch et al., 2018); Vaccine knowledge = Knowledge of COVID-19 vaccine scale; Vulnerable people = Close contact with people known to be vulnerable to COVID-19; SRH = Single-item self-reported health; E = Extroversion personality trait; A = Agreeableness personality trait; C = Conscientiousness personality trait; N = Neuroticism personality trait; O = Openness to experience personality trait; IUS-12 = Intolerance of uncertainty scale short form (Carleton et al., 2007).

\*\*\**p* < .001

\*\**p* < .01

\**p* < .05.

## Concurrent validity

Correlations between VaCCS subscales and sets of socio-political variables, personality and individual difference constructs, and health beliefs and outcomes are presented in Table 7, Table 8 and Table 9, respectively. Associations between VaCCS subscales and concurrent validity measures formed characteristic patterns of associations consistent with expectations.

**Table 6.  Correlations among the Vaccination Concerns in COVID-19 Scales (VaCCS) subscales.**

| Subscale | 1 | 2 | 3 | 4 | 5 | 6 | 7 | 8 |
|---|---|---|---|---|---|---|---|---|
| 1. Efficacy |  | .870 | -.589 | -.510 | -.758 | -.483 | -.577 | .533 |
|  | – | .802 | -.420 | -.430 | -.741 | -.232 | -.512 | .268 |
|  |  | .794 | -.403 | -.411 | -.548 | -.462 | -.553 | .695 |
| 2. Trust | .889 |  | -.673 | -.467 | -.745 | -.571 | -.654 | .592 |
|  | .837 | – | -.486 | -.332 | -.730 | -.299 | -.624 | .348 |
|  | .842 |  | -.436 | -.350 | -.554 | -.470 | -.563 | .733 |
| 3. Worry | -.598 | -.698 |  | .505 | .657 | .766 | .766 | -.533 |
|  | -.425 | -.503 | – | .416 | .506 | .634 | .699 | -.267 |
|  | -.417 | -.454 |  | .641 | .701 | .799 | .684 | -.353 |
| 4. Cause | -.550 | -.518 | .538 |  | .648 | .475 | .476 | -.380 |
|  | -.455 | -.366 | .436 | – | .557 | .286 | .350 | -.160 |
|  | -.461 | -.432 | .678 |  | .771 | .689 | .674 | -.255 |
| 5. Scepticism | -.777 | -.766 | .679 | .684 |  | .563 | .660 | -.490 |
|  | -.748 | -.743 | .509 | .585 | – | .339 | .624 | -.291 |
|  | -.587 | -.605 | .715 | .834 |  | .775 | .793 | -.441 |
| 6. Fear | -.511 | -.606 | .787 | .539 | .587 |  | .715 | -.554 |
|  | -.248 | -.312 | .647 | .327 | .366 | – | .537 | -.262 |
|  | -.491 | -.514 | .812 | .755 | .830 |  | .736 | -.376 |
| 7. Uncertainty | -.620 | -.714 | .821 | .534 | .724 | .750 |  | -.609 |
|  | -.559 | -.689 | .758 | .395 | .687 | .576 | – | -.374 |
|  | -.630 | -.648 | .743 | .774 | .900 | .838 |  | -.521 |
| 8. Literacy | .606 | .646 | -.540 | -.392 | -.520 | -.549 | -.630 |  |
|  | .346 | .425 | -.296 | -.202 | -.336 | -.286 | -.441 | – |
|  | .770 | .790 | -.368 | -.313 | -.495 | -.420 | -.604 |  |

*Note.* The matrix presented above the principal diagonal comprises manifest (averaged) variable correlations, and the matrix below the principal diagonal comprises latent variable correlations. Correlations presented on the upper, center, and lower lines are for the Australian, USA (unvaccinated), and USA (vaccinated) samples, respectively. VaCCS = Vaccination Concerns in COVID-19 Scale; Efficacy = Beliefs in efficacy and prevention VaCCS subscale; Trust = Trust in authorities VaCCS subscale; Worry = Worry about safety and side effects VaCCS subscale; Cause = Beliefs vaccine causes COVID-19 VaCCS subscale; Scepticism = Scepticism and mistrust in the vaccine VaCCS subscale; Fear = Fear of vaccine VaCCS subscale; Uncertainty = Uncertainty and hesitation getting vaccinated VaCCS subscale; Literacy = Vaccine literacy VaCCS subscale.

All correlations are statistically significant ($p < .001$).

**Socio-political beliefs.**  Focusing on correlations with socio-political beliefs, the VaCCS efficacy, trust, and literacy subscales were statistically significantly and positively correlated with the Oxford vaccine hesitancy scale and government trust in all three samples with medium-to-large effect sizes. These scales were also negatively correlated with COVID-19, vaccine, and general conspiracy beliefs, and vaccine hesitancy with medium-to-large effect sizes, although effects for the literacy subscale were smaller. In keeping with this pattern, the VaCCS worry, cause, scepticism, fear, and uncertainty subscales were negatively correlated with the Oxford scale and government trust, and positively related to all conspiracy beliefs variables and vaccine hesitancy, again with medium-to-large effect sizes and in all samples. The efficacy, trust, and literacy subscales were also positively correlated with free will, but only in the Australia and vaccinated USA samples. In addition, the efficacy and trust subscales were negatively correlated with vaccine denial in all samples, and negatively correlated with political trust except in the vaccinated USA sample, all with small effect sizes. The worry, cause, scepticism, fear, and uncertainty subscales were positively related to political trust and vaccine hesitancy in all samples with small effect sizes.

**Table 7. Correlations of Vaccine Concerns in COVID-19 Scale (VaCCS) subscales with socio-political beliefs.**

| Scale | VaCCS subscales | | | | | | | |
|---|---|---|---|---|---|---|---|---|
| | **Efficacy** | **Trust** | **Worry** | **Cause** | **Sceptic.** | **Fear** | **Uncertain.** | **Literacy** |
| Oxford scale | .829*** | .800*** | -.665*** | -.506*** | -.726*** | -.574*** | -.675*** | .513*** |
| | .780*** | .760*** | -.453*** | -.357*** | -.678*** | -.303*** | -.572*** | .270*** |
| | .741*** | .719*** | -.495*** | -.434*** | -.550*** | -.540*** | -.561*** | .594*** |
| Gov. trust | .618*** | .676*** | -.344*** | -.238*** | -.477*** | -.274*** | -.337*** | .384*** |
| | .683*** | .737*** | -.305*** | -.247*** | -.585*** | -.161* | -.446*** | .279*** |
| | .619*** | .721*** | -.252*** | -.207*** | -.351*** | -.287*** | -.343*** | .577*** |
| Polit. orient. | -.159** | -.130 | .141* | .212*** | .222*** | .102 | .140* | -.052 |
| | -.237*** | -.236*** | .158* | .019 | .250*** | .063 | .193** | -.032 |
| | -.217*** | -.252*** | .333*** | .363*** | .425*** | .324*** | .362*** | -.154* |
| Free will | .195*** | .190*** | -.028 | -.027 | -.076 | -.056 | -.043 | .219*** |
| | .042 | .027 | .103 | .170** | .080 | .057 | .137 | .049 |
| | .295*** | .254*** | -.040 | -.002 | -.001 | -.045*** | -.033 | .267*** |
| Polit. trust | -.153* | -.218*** | .169** | .132* | .274*** | .163** | .228*** | -.123 |
| | -.371*** | -.430*** | .199*** | .177** | .442*** | .138 | .327*** | -.097 |
| | .035 | -.068 | .197*** | .183** | .234*** | .154* | .193** | -.010 |
| COVID-19 CB | -.642*** | -.611*** | .473*** | .581*** | .760*** | .408*** | .472*** | -.360*** |
| | -.650*** | -.612*** | .282*** | .383*** | .704*** | .162* | .386*** | -.104 |
| | -.515*** | -.533*** | .496*** | .579*** | .745*** | .566*** | .633*** | -.460*** |
| Vaccine CB | -.681*** | -.670*** | .566*** | .613*** | .828*** | .510*** | .577*** | -.450*** |
| | -.675*** | -.679*** | .453*** | .502*** | .810*** | .299*** | .552*** | -.213*** |
| | -.537*** | -.547*** | .633*** | .710*** | .851*** | .688*** | .772*** | -.456*** |
| General CB | -.458*** | -.504*** | .445*** | .406*** | .538*** | .368*** | .457*** | -.321*** |
| | -.474*** | -.537*** | .334*** | .280*** | .544*** | .215*** | .450*** | -.156* |
| | -.277*** | -.363*** | .391*** | .356*** | .464*** | .398*** | .469*** | -.265*** |
| Vaccine hesitancy | -.405*** | -.480*** | .517*** | .332*** | .465*** | .577*** | .578*** | -.475*** |
| | -.255*** | -.280*** | .383*** | .142* | .299*** | .315*** | .360*** | -.161* |
| | -.337*** | -.320*** | .488*** | .495*** | .543*** | .524*** | .525*** | -.254*** |
| Vaccine denial | -.354*** | -.331*** | .221*** | .159** | .327*** | .189*** | .176** | -.116 |
| | -.257*** | -.322*** | .217*** | .158* | .297*** | .124 | .282*** | -.101 |
| | -.180** | -.176** | .297*** | .258*** | .277*** | .342*** | .314*** | -.104 |

*Note*. Correlations presented on the upper, center, and lower lines are for the Australian, USA (unvaccinated), and USA (vaccinated) samples, respectively. *p*-values are adjusted for multiple tests. VaCCS = Vaccination Concerns in COVID-19 Scale; Efficacy = Beliefs in efficacy and prevention VaCCS subscale; Trust = Trust in authorities VaCCS subscale; Worry = Worry about safety and side effects VaCCS subscale; Cause = Beliefs vaccine causes COVID-19 VaCCS subscale; Scepticism = Scepticism and mistrust in the vaccine VaCCS subscale; Fear = Fear of vaccine VaCCS subscale; Uncertainty = Uncertainty and hesitation getting vaccinated VaCCS subscale; Literacy = Vaccine literacy VaCCS subscale; Oxford scale = Oxford COVID-19 Vaccine Hesitancy Scale (Freeman et al., 2021); Gov. trust = Trust in government organizations in handling COVID-19 (Grimmelikhuijsen & Knies, 2017); Polit. orient. = Political orientation (Kroh, 2007); Free will = Free will beliefs; Polit. trust = Trust in politicians handling of the COVID-19 pandemic; COVID-19 CB = COVID-19 conspiracy beliefs scale (Imhoff & Lamberty 2020); Vaccine CB = Vaccine conspiracy beliefs scale (Shapiro et al., 2016); General CB = General conspiracy beliefs scale (Bruder et al., 2013); Vaccine hesitancy = Single-item measure of vaccine hesitancy; Vaccine denial = Single-item measure of vaccine denial.

\*\*\**p* < .001

\*\**p* < .01

\**p* < .05.

**Personality and individual difference constructs.** No clear pattern of statistically significant correlations emerged between the VaCCS subscales and the personality and individual difference constructs across the samples. Across samples, only the fear subscale was positively

**Table 8. Correlations of Vaccine Concerns in COVID-19 Scale (VaCCS) subscales with personality and traits.**

| Scale | VaCCS subscales | | | | | | | |
|---|---|---|---|---|---|---|---|---|
| | Efficacy | Trust | Worry | Cause | Sceptic. | Fear | Uncertain. | Literacy |
| Extroversion | -.073 | -.080 | .069 | .107 | .090 | .067 | .012 | -.043 |
| | .009 | -.023 | -.108 | -.052 | -.017 | -.112 | -.064 | .020 |
| | .090 | .039 | -.083 | .012 | .000 | -.086 | -.036 | .058 |
| Agreeableness | .187** | .152* | -.053 | -.149* | -.187** | -.032 | -.048 | .100 |
| | .053 | .041 | .017 | -.113 | -.071 | .037 | .012 | .035 |
| | .068 | .111 | -.108 | -.227*** | -.234*** | -.178** | -.250*** | .106 |
| Conscientiousness | .128 | .072 | -.014 | -.140 | -.169** | -.059 | -.026 | .128 |
| | -.003 | -.076 | .093 | -.140 | -.004 | .072 | .145 | .054 |
| | .160* | .112 | -.324*** | -.366*** | -.385*** | -.404*** | -.351*** | .193** |
| Neuroticism | -.028 | -.009 | .008 | .073 | .066 | .100 | .048 | -.139 |
| | -.039 | -.007 | .061 | .108 | .065 | .178** | .062 | -.066 |
| | -.090 | -.125 | .202*** | .170** | .188** | .303*** | .219*** | -.129 |
| Openness to experience | .144* | .127 | -.036 | -.094 | -.110 | -.082 | -.111 | .144* |
| | .079 | .062 | -.030 | -.008 | -.071 | -.056 | -.018 | .043 |
| | .176** | .208*** | -.127 | -.117 | -.181** | -.205*** | -.172** | .182** |
| IUS-12 | -.034 | -.044 | .107 | .101 | .123 | .176** | .135 | -.214*** |
| | .036 | .018 | .120 | .173** | .049 | .230*** | .124 | -.108 |
| | -.038 | -.058 | .152* | .146* | .191** | .232*** | .173** | -.085 |

*Note.* Correlations presented on the upper, center, and lower lines are for the Australian, USA (unvaccinated), and USA (vaccinated) samples, respectively. *p*-values are adjusted for multiple tests. VaCCS = Vaccination Concerns in COVID-19 Scale; Efficacy = Beliefs in efficacy and prevention VaCCS subscale; Trust = Trust in authorities VaCCS subscale; Worry = Worry about safety and side effects VaCCS subscale; Cause = Beliefs vaccine causes COVID-19 VaCCS subscale; Scepticism = Scepticism and mistrust in the vaccine VaCCS subscale; Fear = Fear of vaccine VaCCS subscale; Uncertainty = Uncertainty and hesitation getting vaccinated VaCCS subscale; Literacy = Vaccine literacy VaCCS subscale; IUS-12 = Intolerance of uncertainty scale short form (Carleton et al., 2007).

***$p < .001$

**$p < .01$

*$p < .05$.

associated with intolerance of uncertainty in all three samples with small effect sizes. The only other patterns of correlations were confined to specific samples. The VaCCS worry, cause, scepticism, fear, and uncertainty subscales were negatively correlated with conscientiousness and agreeableness, and positively related to neuroticism, in the USA (vaccinated) sample with small-to-medium effect sizes.

**Health beliefs and outcomes.** Turning to correlations with health beliefs and outcomes, the VaCCS efficacy, trust, and literacy subscales were statistically significantly and positively correlated with intentions to get the vaccine, and negatively related to risk perceptions, beliefs about medicines, and vaccine confidence with medium-to-large sized effects in all samples. Analogously, the VaCCS worry, cause, scepticism, fear, and uncertainty subscales were negatively correlated with intentions, and positively correlated with risk perceptions, beliefs about medicines, and vaccine confidence with medium-to-large sized effects and in all samples. The VaCCS efficacy and trust subscales were positively correlated, and the VaCCS worry, cause, scepticism, fear, and uncertainty subscales negatively correlated, with vaccine knowledge except in the vaccinated USA sample. A similar pattern of correlations was exhibited with receiving the influenza vaccine and influenza vaccine regularity, but only in the Australia sample. The VaCCS efficacy and trust subscales were positively correlated with receiving an influenza vaccine and influenza vaccine regularity in all three samples.

**Table 9. Correlations of Vaccine Concerns in COVID-19 Scale (VaCCS) subscales with health-related beliefs and outcomes.**

| Scale | VaCCS subscales | | | | | | | |
|---|---|---|---|---|---|---|---|---|
| | Efficacy | Trust | Worry | Cause | Sceptic. | Fear | Uncertain. | Literacy |
| Intention | .807*** | .803*** | -.673*** | -.476*** | -.683*** | -.571*** | -.684*** | .536*** |
| | .701*** | .716*** | -.416*** | -.280*** | -.600*** | -.300*** | -.548*** | .278*** |
| | .769*** | .747*** | -.462*** | -.390*** | -.518*** | -.506*** | -.514*** | .651*** |
| Risk perceptions | -.589*** | -.641*** | .712*** | .469*** | .631*** | .692*** | .678*** | -.555*** |
| | -.569*** | -.576*** | .622*** | .350*** | .578*** | .479*** | .614*** | -.283*** |
| | -.459*** | -.471*** | .697*** | .669*** | .753*** | .768*** | .707*** | -.379*** |
| BMQ | -.458*** | -.426*** | .443*** | .509*** | .562*** | .424*** | .435*** | -.315*** |
| | -.361*** | -.338*** | .281*** | .381*** | .473*** | .204*** | .342*** | -.149* |
| | -.282*** | -.238*** | .459*** | .561*** | .615*** | .540*** | .581*** | -.202*** |
| Vaccine confidence[a] | -.476*** | -.425*** | .440*** | .510*** | .585*** | .442*** | .404*** | -.299*** |
| | -.293*** | -.262*** | .333*** | .333*** | .423*** | .307*** | .373*** | -.187** |
| | -.425*** | -.379*** | .576*** | .613*** | .739*** | .656*** | .676*** | -.328*** |
| Vaccine knowledge | .300*** | .267*** | -.286*** | -.329*** | -.358*** | -.283*** | -.267*** | .291*** |
| | .048 | .096 | -.073 | -.149* | -.118 | -.109 | -.156* | .221*** |
| | .360*** | .360*** | -.272*** | -.263*** | -.303*** | -.282*** | -.348*** | .389*** |
| Vulnerable people | -.107 | -.104 | -.001 | -.007 | .050 | -.062 | -.010 | .005 |
| | -.128 | -.082 | -.031 | .071 | .089 | -.103 | .036 | .002 |
| | .007 | .019 | -.173** | -.141 | -.148 | -.146 | -.148 | -.010 |
| SRH | .135 | .123 | -.075 | -.070 | -.047 | -.094 | -.082 | .149* |
| | -.059 | -.069 | -.023 | .047 | .103 | -.124 | .043 | .061 |
| | .164* | .217*** | -.039 | .048 | .014 | -.051 | -.020 | .236** |
| Flu shot | .357*** | .328*** | -.253*** | -.225*** | -.314*** | -.218*** | -.298*** | .258*** |
| | .145 | .109 | -.070 | -.114 | -.161* | -.040 | -.098 | .016 |
| | .181** | .147 | -.078 | -.045 | -.048 | -.129 | -.111 | .177**s |
| Regular flu shot | .447*** | .411*** | -.323*** | -.267*** | -.358*** | -.268*** | -.347*** | .304*** |
| | .237*** | .215*** | -.092 | -.128 | -.181** | -.077 | -.151* | .058 |
| | .262*** | .219*** | -.079 | -.038 | -.059 | -.129 | -.130 | .218*** |

*Note.*

[a]High scores on this scale represent lower confidence in vaccines. Correlations presented on the upper, center, and lower lines are for the Australian, USA (unvaccinated), and USA (vaccinated) samples, respectively. *p*-values are adjusted for multiple tests. VaCCS = Vaccination Concerns in COVID-19 Scale; Efficacy = Beliefs in efficacy and prevention VaCCS subscale; Trust = Trust in authorities VaCCS subscale; Worry = Worry about safety and side effects VaCCS subscale; Cause = Beliefs vaccine causes COVID-19 VaCCS subscale; Scepticism = Scepticism and mistrust in the vaccine VaCCS subscale; Fear = Fear of vaccine VaCCS subscale; Uncertainty = Uncertainty and hesitation getting vaccinated VaCCS subscale; Literacy = Vaccine literacy VaCCS subscale; Intention = COVID-19 vaccination intentions; Risk perceptions = Beliefs in risks of COVID-19; BMQ = Beliefs about medicines questionnaire; Vaccine confidence = Vaccine confidence scale (Betsch et al., 2018); Vaccine knowledge = Knowledge of COVID-19 vaccine scale; Vulnerable people = Close contact with people known to be vulnerable to COVID-19; SRH = Single-item self-reported health; Flu shot = Received influenza vaccine in the past year; Regular flu shot = Regular recipient of influenza vaccine.

***$p < .001$

**$p < .01$

*$p < .05$.

## Predictive validity

We also examined the unique effects of the VaCCS subscales on intentions to get the COVID-19 vaccine, or, in the case of the USA vaccinated sample, intentions to get a COVID-19 'booster' vaccine, alongside socio-demographic covariates, risk perceptions, and vaccine hesitancy. Results of the linear regression of intentions on the VaCCS subscales, risk perceptions, vaccine hesitancy, and socio-demographic variables in each sample are presented in Table 10. Across the samples,

**Table 10. Results of linear multiple regression analysis of effects of Vaccine Concerns in COVID-19 Scale (VaCCS) subscales, risk perceptions, and socio-demographic variables on vaccine intentions.**

| Variable | Australia sample | | | USA unvaccinated sample | | | USA vaccinated sample | | |
|---|---|---|---|---|---|---|---|---|---|
| | β | 95% CI | | B | 95% CI | | β | 95% CI | |
| | | LB | UB | | LB | UB | | LB | UB |
| Age | .036 | -.009 | .082 | .039 | -.016 | .093 | -.011 | -.069 | .047 |
| Gender | -.030 | -.074 | .014 | -.057* | -.112 | -.001 | -.003 | -.057 | .050 |
| Education | .026 | -.019 | .070 | .019 | -.036 | .074 | .008 | -.046 | .062 |
| Employment status | .007 | -.037 | .051 | -.068* | -.123 | -.012 | .025 | -.029 | .079 |
| Race | -.039 | -.085 | .007 | -.022 | -.077 | .033 | -.004 | -.057 | .049 |
| COVID history | -.047* | -.089 | -.004 | -.019 | -.073 | .036 | -.020 | -.075 | .036 |
| Risk perceptions | -.129*** | -.198 | -.060 | -.221*** | -.300 | -.142 | -.066 | -.156 | .023 |
| Vaccine hesitancy | -.100*** | -.156 | -.045 | -.123*** | -.183 | -.063 | -.117*** | -.180 | -.054 |
| VaCCS Efficacy | .464*** | .374 | .554 | .316*** | .215 | .416 | .410*** | .322 | .498 |
| VaCCS Trust | .179*** | .080 | .278 | .317*** | .214 | .421 | .274*** | .177 | .370 |
| VaCCS Worry | -.098* | -.179 | -.017 | .112* | .023 | .202 | -.067 | -.156 | .022 |
| VaCCS Cause | -.016 | -.073 | .041 | .031 | -.038 | .100 | -.008 | -.095 | .079 |
| VaCCS Scepticism | .069 | -.012 | .150 | .029 | -.071 | .129 | .070 | -.044 | .183 |
| VaCCS Fear | .048 | -.027 | .122 | -.016 | -.090 | .057 | -.114* | -.220 | -.009 |
| VaCCS Uncertainty | -.173*** | -.249 | -.096 | -.093* | -.184 | -.002 | .115* | .019 | .212 |
| VaCCS Literacy | -.044 | -.102 | .014 | .002 | -.057 | .062 | .132** | .052 | .213 |

*Note*. Model $R^2$ values for intention were .762, .621, and .681 for the Australia, USA (vaccinated), and USA (unvaccinated) samples, respectively. β = Standardized regression coefficient; 95% CI = 95% Confidence interval of β; LB = Lower bound of 95% CI; UB = UB of 95% CI; Risk perceptions = Beliefs in risks of COVID-19; Vaccine hesitancy = Single-item measure of vaccine hesitancy; VaCCS = Vaccination Concerns in COVID-19 Scale; Efficacy = Beliefs in efficacy and prevention VaCCS subscale; Trust = Trust in authorities VaCCS subscale; Worry = Worry about safety and side effects VaCCS subscale; Cause = Beliefs vaccine causes COVID-19 VaCCS subscale; Scepticism = Scepticism and mistrust in the vaccine VaCCS subscale; Fear = Fear of vaccine VaCCS subscale; Uncertainty = Uncertainty and hesitation getting vaccinated VaCCS subscale; Literacy = Vaccine literacy VaCCS subscale.

***$p < .001$

**$p < .01$

*$p < .05$.

the VaCCS *efficacy* and *trust* subscales were statistically significant positive predictors of intentions with small-to-medium effect sizes, alongside a significant negative effect of vaccine hesitancy with a small effect size. The VaCCS *uncertainty* subscale was also a significant predictor across the samples, with small effect sizes, although the effect was negative in the unvaccinated samples, but positive in the vaccinated sample. This positive, statistically significant effect was unexpected given the large, statistically significant and negative correlation between intention and the uncertainty subscale ($r = -.514$, $p < .001$; Table 9). This is likely to be a suppressor effect caused by the substantive correlation between the uncertainty subscale and the other VaCCS subscales in this sample, particularly the scepticism and uncertainty subscales (Table 6). Further, risk perception was also a significant negative predictor of intentions in the unvaccinated samples, with a small effect size, but not in the vaccinated sample. In addition, the VaCCS *fear* and *literacy* subscales were significant predictors of intentions in the latter sample. Overall, these constructs accounted for between 62.1% and 76.2% of the variance in vaccine intentions across the samples.

## Discussion

The present study reports on the development and initial validation of the Vaccine Concerns in COVID-19 Scale (VaCCS), a psychometric instrument designed to measure individuals'

beliefs and concerns with respect to COVID-19 vaccines. The measure was developed in a multi-stage procedure from first principles comprising a developmental stage, in which an initial item pool was provided via an evidence synthesis of previous vaccine scales and an open-ended survey, and a validation stage in which a final pool of items was selected via rigorous construct and concurrent validity assessment in samples of Australian and the USA residents. The procedure produced a final 35-item scale with eight subscales. The scale exhibited strong psychometric integrity with a coherent factor structure that was invariant across samples. Subscale scores exhibited a predictable pattern of correlations with salient measures of socio-political beliefs, health beliefs and outcomes, and some selected personality and individual difference constructs. In addition, the VaCCS *efficacy* and *trust* subscales were consistent positive predictors of intentions to get the COVID-19 vaccine across samples, alongside vaccine hesitancy and risk perceptions. In addition, the *uncertainty* VaCCS subscale was negatively associated with COVID-19 vaccine intentions in the unvaccinated samples, but positively associated with intentions in the vaccinated sample.

## A comprehensive measure of COVID-19 vaccine beliefs and concerns

Our development process yielded an instrument that captures the wide-ranging sets of beliefs likely to be of relevance to individuals' decisions to get the COVID-19 vaccine. Concerns about the vaccine featured prominently in the scale and were captured by a number of its subscales. For example, the *worry* and *uncertainty* subscales reflect safety concerns regarding the rapid development of the vaccines and a lack of long-term studies of their effects, and concerns that the vaccine may lead to adverse side effects. These concerns have also been identified in research examining safety concerns in the context of COVID-19 [74, 75]. In addition, the *fear* subscale captures anxiety and fear over getting a vaccine, which may also be linked with safety concerns, or stem from a generalized fear of medicines or medical procedures, or with the process of vaccine administration such as a fear of injections or syringes [76]. Further concerns are captured by the *scepticism* and *trust* subscales. The *trust* subscale taps into individuals' general mistrust of the government, scientists, and pharmaceutical corporations responsible for developing and administering the vaccines, and likely captures generalized "anti-vax" beliefs. Similarly, vaccines have been equated as an instrument of governmental control, particularly with recent mandates that require vaccinations among essential workers [77–79], beliefs captured by the *scepticism* subscale. There have also been concerns over the efficacy of the vaccines, particularly in light of highly-publicized, albeit rare, cases where vaccinated individuals have been hospitalized with severe COVID-19, and reports of high infection rates among the vaccinated as new, highly contagious variants of the virus spread. The *efficacy* subscale, in particular, captures these beliefs, particularly concerns that vaccines may not be sufficiently effective or may do more harm to health than good. There is also evidence that individuals may harbor beliefs focused on the vaccine itself and that it may infect individuals with COVID-19 [80, 81]–the *cause* subscale taps into these beliefs. Such beliefs likely stem from a lack of understanding of the vaccine and how it works, and a perceived lack of clear information on the vaccine and its effects, both of which may be related to low levels of health literacy. Such beliefs identified in the *literacy* subscale. Taken together, the VaCCS represents a comprehensive measure that captures a range of beliefs individuals may hold with respect to getting a COVID-19 vaccine, and are likely to be implicated in their future decisions to get the vaccine.

## Concurrent validity

Given the breadth of beliefs captured by the VaCCS, examination of relations between the subscales and measures of conceptually-related beliefs and constructs was important to provide

evidence for their concurrent validity. To this end, we administered measures of beliefs and perceptions expected to reflect the potential origins or consequences of beliefs and concerns about vaccines alongside the new measure.

**Socio-political beliefs.** Taking into consideration the social and political context in which the COVID-19 pandemic has occurred and developed, a prominent set of beliefs with which subscales of the VaCCS subscales were expected to be related was socio-political beliefs [37, 38]. Confirming our expectations, we observed statistically-significant correlations between key VaCCS subscales and beliefs that represent lack of trust in the government, beliefs in governmental and broader conspiracy theories, and vaccine hesitancy. Individuals who scored higher on the *efficacy* and *trust* subscales were more likely to report high general trust in government and other organizations, and less likely to express vaccine hesitancy or denial, or to endorse conspiracy beliefs about the vaccine. Individuals endorsing the *efficacy* and *trust* subscales were also more likely to identify as liberal or politically left-wing. In contrast, those who scored lower on the *efficacy* and *trust* subscales, and reported higher levels on the *scepticism*, *uncertainty*, *worry*, and *fear* subscales, were more likely to report low trust in government and vaccine producers, be vaccine hesitant or be in denial about the pandemic, endorse conspiracy theories, and identifying as more conservative or right-wing political ideology. These patterns of associations were highly consistent across the three samples, suggesting that they were not a symptom of the localized political climate. Taken together, this pattern of associations corroborates research linking concerns, mistrust, and scepticism with reduced likelihood of endorsing the vaccine among those expressing conservative views [37, 38, 79]. The findings are also consistent with high-profile reporters', politicians' and other 'celebrity' figures' expressions of mistrust of the vaccines, and promulgation of conspiracy theories without credible evidence, via the populist press and social media [82, 83]. These effects provide evidence supporting the concurrent validity of the VaCCS subscales with respect to the political landscape and dominant views that pervade in during the roll-out of the vaccines.

Interestingly, we observed few relations between the VaCCS subscales and free will beliefs. While we expected that individuals' concerns that the vaccine represents government interference and overreach, the VaCCS *trust* and *uncertainty* subscales were generally not correlated with free will beliefs. One possibility is that free will is associated with attitudes towards vaccine mandates, such as proof of vaccine requirements for travel, work, and entry into bars and restaurants, rather than to the vaccine itself, although these beliefs would be expected to correlate. Alternatively, the lack of associations might be because free will beliefs are highly generalized and focus on global beliefs about capacity and agency, rather than beliefs in specific governmental and organization impingements that might affect specific decisions such as getting the COVID-19 vaccine. The same may apply to relations between the specific beliefs captured by the VaCCS subscales and more generalized views captured in trait-level constructs, which we discuss next.

**Personality and individual difference constructs.** We found relatively few statistically significant correlations between the VaCCS subscales and the personality and individual difference constructs. Subscales representing concerns about the vaccine, particularly the *cause*, *scepticism*, *fear*, and *uncertainty* tended to be negatively related to the agreeableness and conscientiousness personality constructs, and negatively related to neuroticism, in the vaccinated sample but not in the other samples. This suggests that individuals with tendencies toward harmonious social relations and work ethic and organization were less likely to harbor concerns about the vaccine, and were more certain of its credibility. However, the most consistent finding was the positive association between the *fear* subscale and intolerance for uncertainty across samples. Uncertainty over the effects of the vaccine is mirrored by the level of fear individuals' express over getting the vaccine. This finding is consistent with theory [84] and

research [85, 86] demonstrating that uncertainty is likely to generate feelings of fear in health contexts, and perhaps fear of COVID itself although this needs empirical corroboration, and provides further evidence for the concurrent validity of this subscale. The absence of clear trends in correlations between VaCCS subscales and traits may be a function of the generalized focus of the traits compared to the VaCCS subscales which refer to a specific behavioral context. Traits are expected to be associated with beliefs and behavioral tendencies across a wide range of behaviors and contexts, albeit with relatively small effect sizes, consistent with the trends observed here. Overall, the largely trivial correlations observed with the personality and individual difference constructs do not add substantively to evidence in support of the concurrent validity of the VaCCS.

**Health beliefs and outcomes.** The VaCCS subscales were also expected to be consistently related to health beliefs and outcomes, particularly generalized beliefs about medicines, knowledge about vaccines, beliefs about COVID-19 risks, the vulnerability of others around them to infection, and individuals' past vaccination behavior including the tendency to get an influenza vaccination. Specifically, the *efficacy* and *trust* VaCCS subscales were positively related to vaccine knowledge and past history of getting an influenza vaccine, and negatively related to vaccine risk perceptions, concerns about medicines, and lack of confidence in vaccines. A similar characteristic pattern was observed for the VaCCS subscales relating to *worry* and *fear* about the vaccine, and *scepticism* and *uncertainty* of the government and vaccine producers, with those scoring highly on these subscales more likely to report lower knowledge about the vaccine, view the vaccine as risky, express concerns about medicines, and be less likely to have received an influenza vaccine or regularly get a flu shot. This pattern of effects may be attributable to low generalized health literacy and knowledge. The *literacy* subscale was consistently and positively related to knowledge about the vaccine, and negatively related to VaCCS subscales relating to concerns and mistrust. This was also corroborated by the positive association between the *literacy* subscale and vaccine hesitancy, lack of trust in the government, and endorsement of conspiracy beliefs, suggesting that individuals who have generally low ability to take on and interpret information about COVID-19 vaccines may be more vulnerable to misinformation. These findings are consistent with research indicating that individuals' beliefs about the health concerns and risks with respect to the COVID-19 vaccine are consistent with their generalized health concerns and vaccine behavior [87–89], and, most importantly their motivation to get vaccinated [90, 91]. These beliefs may also be associated with perceptions of risk and fear of COVID-19. However, these relationships were not tested in the current analysis, and we look to future studies to provide empirical verification. Taken together, this pattern of effects provides further support for the VaCCS as an instrument that yields a profile of beliefs with respect to the COVID-19 vaccine that are consistently with related to the health-related beliefs that likely impact future behavior and to motives to get vaccinated.

## Prediction of COVID-19 vaccine intention

The VaCCS subscales were also expected to be implicated in individuals' motivation to get the COVID-19 vaccine in future, or, in the case of the vaccinated sample, the booster vaccine. This was corroborated by positive associations between the *efficacy* and *trust* subscales with vaccine intentions, and negative associations with subscales reflecting *concerns*, *scepticism*, and *mistrust*. Also important, however, was our regression analysis, which provided evidence on the VaCCS subscales that contributed most to explaining variance in intentions. Our analysis also examined effects of subscales on intentions concurrent with other beliefs likely to be implicated in vaccine decision-making, namely, risk perceptions and vaccine hesitancy, and controlled for effect of salient demographic covariates. Focusing on the prediction of vaccine

intentions in the unvaccinated samples, the *efficacy* and *trust* subscales, along with risk perceptions, were the most prominent predictors, while the *uncertainty* and *worry* subscales, along with vaccine hesitancy, had much smaller effects. In contrast, effects of the *cause*, *scepticism*, *fear*, and *literacy* subscales were trivial and not statistically significant. These findings corroborate previous research demonstrating the prominence of risk perceptions, and beliefs in the efficacy of the vaccine and trust in government and vaccine producers, as driving individuals' intentions to get vaccinated [32, 33, 92, 93], and provides important predictive validity for the VaCCS subscales. The pattern of prediction was similar for the prediction of vaccine booster intentions among the vaccinated sample, with *efficacy* and *trust*, along with vaccine hesitancy prominent statistically significant predictors, with smaller the effects of the *fear*, *uncertainty*, and *literacy* subscale. This slightly different pattern of effects for booster intentions indicates the continued importance of endorsing vaccine efficacy and trust, and low vaccine hesitancy, in determining vaccine-related decision making going forward. Findings may also have resonance when it comes to informing messaging and interventions aimed at promoting vaccination uptake. Interventionists may consider developing communications that emphasize the efficacy and trustworthiness of the vaccine, particularly its effectiveness in preventing serious COVID-19 infections that require hospitalization and in reducing the spread of infections, and highlight the robustness of data supporting its safety and the credibility of the pharmaceutical companies that have developed the vaccine.

## Strengths, limitations, and avenues for future research

We aimed to develop a comprehensive psychometric instrument to capture individuals' beliefs and concerns with respect to the COVID-19 vaccine. The scale was developed and validated from first principles in a multi-stage approach with numerous design and methodological strengths, including: (a) identification of candidate beliefs from a review of prior vaccine research and the views and perspectives of individuals eligible to receive the vaccine, the target audience; (b) formation of candidate items through content analysis of candidate beliefs from both sources; (c) rigorous factor analytic validation in multiple samples of residents from two countries during a time when COVID-19 vaccines were being rolled out; and (d) concurrent and predictive validity tests through associations with key socio-political beliefs, health beliefs and outcomes, and personality and individual difference constructs and the prediction of vaccine intentions. While current findings provide good evidence in support of the structural integrity, internal consistency, and validity of the VaCCS, findings should be interpreted in light of a number of limitations. These include the exclusive reliance on correlational data and self-report measures, the lack of a behavioral measure, the non-representativeness of the samples used in the validation studies, and the possibility that the scale does not encompass all beliefs and concerns with respect to getting a COVID-19 vaccine. Next, we outline each of these limitations, and provide some suggestions for future research.

Data used in the validation process of the VaCCS were correlational, and the measures adopted were exclusively self-report. Correlational data are not informative on causal effects and are not able to identify possible 'third variable' explanations for correlations. For example, we cannot, on the basis of the current data, conclude that the sets of beliefs captured by the VaCCS are causally related to outcomes such as vaccine intentions or hesitancy. In addition, the current data did not enable us to ascertain the sensitivity and specificity of the VaCCS subscales in measuring belief change, such as belief change that may be brought about by the advent of new information or the introduction of persuasive communications about the COVID-19 vaccines. A further limitation is the reliance on self-report measures, particularly for the measures used in concurrent validity tests, which may introduce error variance to the

correlations due to recall and reporting biases. A related limitation is the lack of a behavioral measure of vaccine uptake, which would provide further evidence in support for the predictive validity of the scale. Resolution lies in experimental or intervention research that tests the effects of strategies to alter or manipulate vaccine beliefs and concerns on change in the VaCCS subscales. It would also be important to examine change in VaCCS subscales prior to and after vaccination, or the association between the subscales and subsequent vaccination status over time. Such research will test the value of the VaCCS in capturing change in beliefs and concerns, and its association with vaccine behavior.

A further limitation is that the samples used in the validation phase of the current research were not representative of the Australian or USA populations from which they were drawn. While participants from the current samples generally reflected the population in terms of age and gender distribution, participants were predominantly white, from higher income and education backgrounds, and more likely to be employed. Given research suggesting that COVID-19 infection and vaccination rates differ greatly across different demographic groups, with those from minority race and ethnic groups and those on low incomes or from underserved communities more likely to be infected and have more serious consequences [94–96], and less likely to be vaccinated [37], research replicating current findings in more representative samples, or making explicit comparisons across demographic groups is needed to provide further support for the validity of the VaCCS. It is also important for research to replicate current findings in non-English speaking and under-resourced countries to ensure generalizability of the VaCCS more broadly.

Finally, the VaCCS may not fully encompass all beliefs with respect to COVID-19 vaccinations. The current scale was designed to be comprehensive insofar as it captured the COVID-19 vaccine beliefs and concerns held by the majority of participants in the current samples, and those that have been reflected in previous research on vaccines [97]. However, this does not rule out the possibility that beliefs and concerns salient to specific populations, or idiosyncratic beliefs and concerns relevant to smaller segments of the population, such as religious groups, exist that were not identified in the development stages, or failed to emerge in our factor analyses. For example, beliefs about healthcare access and treatment costs did not emerge from the current analysis. This may be because participants were sourced from countries where vaccine access is assumed to be provided free of charge, or that participants were largely from high-income groups where healthcare access is not a primary consideration. We look to future formative research to examine whether such beliefs may be salient considerations with respect to the COVID-19 vaccine in other national groups, or in groups from underserved communities.

## Conclusion

Given the essential role that vaccines play in reducing serious COVID-19 cases and in reducing infection spread, the goal of the current study was to provide researchers, professionals, and policymakers working in healthcare contexts and COVID-19 vaccine vaccination programs with a valid and reliable measure that captures the beliefs likely to be implicated in individuals' decisions to get vaccinated. Our scale was developed from first principles in a rigorous multi-stage process, and the emergent 36-item, 8-subscale measure demonstrated good psychometric integrity, and concurrent and predictive validity. The current research paves the way for the research community to now apply the scale as a means to assess the strength of beliefs and concerns with respect to the COVID-19 vaccine, and used it as a means to assess candidate correlates of vaccine intentions and behavior. Such research will further contribute to the evidence base for the validity of the scale, and establish evidence for its broader application in diverse populations and contexts.

## Supporting information

**S1 File. Coding of participant responses and theme extraction in Phase 1 of the study.**
(DOCX)

**S2 File. Search syntax.**
(DOCX)

**S3 File. PRISMA flow chart.**
(DOCX)

**S4 File. Extracted scale items and their sources.**
(DOCX)

## Acknowledgments

We thank Daniel Phipps for his assistance with literature search and data collection and extraction.

## Author Contributions

**Conceptualization:** Kyra Hamilton, Martin S. Hagger.

**Data curation:** Kyra Hamilton, Martin S. Hagger.

**Formal analysis:** Kyra Hamilton, Martin S. Hagger.

**Methodology:** Kyra Hamilton, Martin S. Hagger.

**Project administration:** Kyra Hamilton, Martin S. Hagger.

**Validation:** Kyra Hamilton.

**Writing – original draft:** Kyra Hamilton, Martin S. Hagger.

**Writing – review & editing:** Kyra Hamilton, Martin S. Hagger.

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
