## [Decision Letter · Decision Letter 0]

17 Jan 2022

PONE-D-21-37977The Vaccination Concerns in COVID-19 Scale (VaCCS): Development and ValidationPLOS ONE

Dear Dr. Hagger,

Thank you for submitting your manuscript to PLOS ONE. After careful consideration, we feel that it has merit but does not fully meet PLOS ONE’s publication criteria as it currently stands. Therefore, we invite you to submit a revised version of the manuscript that addresses the points raised during the review process. In the revised version of the paper, please try to provide more insight regarding the choices you have made along the paper in the modelling process. Please consider the reviewers' comments listed at the bottom of this email.

We look forward to receiving your revised manuscript.

Kind regards,

Camelia Delcea

Academic Editor

PLOS ONE

Journal Requirements:

Reviewers' comments:

Reviewer's Responses to Questions

**Comments to the Author**

1. Is the manuscript technically sound, and do the data support the conclusions?

Reviewer #1: Partly

Reviewer #2: Yes

2. Has the statistical analysis been performed appropriately and rigorously? 

Reviewer #1: No

Reviewer #2: Yes

3. Have the authors made all data underlying the findings in their manuscript fully available?

Reviewer #1: No

Reviewer #2: No

4. Is the manuscript presented in an intelligible fashion and written in standard English?

Reviewer #1: Yes

Reviewer #2: Yes

5. Review Comments to the Author

Reviewer #1: The study reports on the development and initial validation of the Vaccine Concerns in COVID-19 Scale (VaCCS), a psychometric instrument designed to measure individuals’ beliefs and concerns with respect to COVID-19 vaccines.

Thank you for the opportunity to review the paper. I found it very interesting and timely, however I have a few concerns:

Major

I found the “Participants and Recruitment” section quite confusing and fairly unstructured. Here a few examples:

• The authors mention different phases of scale points, without defining them or providing the reader with the differences among them.

• I found similar issues in the abstract, but I totally understand that with a strict limit word the authors might not be able to extensively explain the differences between the phases of scale, so I would recommend them to clarify those in the main article content.

• Initially the authors recruit only unvaccinated individuals, but then a sample of (fully/partially?) vaccinated individuals is recruited for the USA. I cannot understand the logic behind such a decision. Could the authors provide the reader with some explanation for that?

It is unclear to me, how this paper contributes to the literature. Is it through the creation and validation of a serious of indicators which might explain Vaccination Concerns in COVID-19 Scale? I would recommend the authors to highlight the contribution and state it (or their research question clearly).

Minor

• The choice of the items for the VaCCS are backed-up by very few references, which appear quite surprising considering that the authors have done a systematic review to choose them. I would suggest them to provide a few more reference so they can bolster their methods.

Reviewer #2: This is a very timely and important study. The methods are well detailed and robust. I have made some suggestions for improvement in the attached pdf as comments. My main suggestion is that the methods are redundant in some places (the same thing is explained several times), and several sections could be consolidated.

6. PLOS authors have the option to publish the peer review history of their article (what does this mean?). If published, this will include your full peer review and any attached files.

Reviewer #1: No

Reviewer #2: **Yes: **Ariadna Capasso

---

## [Author Response · Author response to Decision Letter 0]

2 Feb 2022

Camelia Delcea

Academic Editor

PLOS ONE

Date: February 2, 2022

Manuscript Number: PONE-D-21-37977

The Vaccination Concerns in COVID-19 Scale (VaCCS): Development and Validation

Dear Dr. Delcea,

We would like to thank you and reviewers for taking the time to review our manuscript: “The Vaccination Concerns in COVID-19 Scale (VaCCS): Development and Validation”, which we submitted for consideration for publication in PLOS ONE. We are very grateful for the interest in our paper and appreciate the opportunity for it to be further considered for publication. We have endeavored to make the changes suggested (indicated alongside the reviewers’ original comments below and in tracked changes in the revised manuscript). The feedback has greatly enhanced the quality of our manuscript.

REVIEWER 1 

REVIEWER COMMENT: The study reports on the development and initial validation of the Vaccine Concerns in COVID-19 Scale (VaCCS), a psychometric instrument designed to measure individuals’ beliefs and concerns with respect to COVID-19 vaccines. Thank you for the opportunity to review the paper. I found it very interesting and timely, however I have a few concerns.

AUTHORS’ RESPONSE: We thank Reviewer 1 for their positive feedback, and for taking the time to review our manuscript. We have aimed to address each of their comments below, and also highlight the changes we have made in the manuscript.

REVIEWER COMMENT: The authors mention different phases of scale points, without defining them or providing the reader with the differences among them.

AUTHORS’ RESPONSE: Thank you for this comment. We agree that description of the different phases of the scale development process could be improved. We have now clarified the different phases in both the ‘Participant and Recruitment’ and ‘Design and Procedure’ sections of the Methods. Specifically, we state that we followed a systematic, multi-step design process using the AMEE Guide to develop the VaCCS. The AMEE Guide presents a seven-step survey scale design process broadly consisting of a development phase (steps 1-6) and a validation phase (step 7). In the ‘Design and Procedure’ section, we have now also outlined and described each of these steps.

REVIEWER COMMENT: I found similar issues in the abstract, but I totally understand that with a strict limit word the authors might not be able to extensively explain the differences between the phases of scale, so I would recommend them to clarify those in the main article content.

AUTHORS’ RESPONSE: Thank you for this understanding, we have made minor amendments to the Abstract, such as making reference to the AMEE Guide used to develop the VaCCS. We anticipate this will give readers indication of the systematic, muti-step process that we used to develop and validate the VaCCS.

REVIEWER COMMENT: Initially the authors recruit only unvaccinated individuals, but then a sample of (fully/partially?) vaccinated individuals is recruited for the USA. I cannot understand the logic behind such a decision. Could the authors provide the reader with some explanation for that?

AUTHORS’ RESPONSE: The recruitment of the additional sample of vaccinated USA residents was used to replicate the validation procedures. We have now added this point in the revised paper. “We also collected data from an additional sample of vaccinated USA residents (N = 479, 56.8% female) between June 3 and June 7, 2021, which we used to replicate the validation procedures.” We thought this additional sample and analysis was important in order to provide evidence of the concerns and beliefs that vaccinated people hold given the introduction of COVID-19 booster shots and the likely necessity of needing further ‘booster’ or even yearly COVID-19 shots in future. Such information can signpost potentially modifiable targets for behavioral interventions aimed at fostering uptake of booster shots and potentially continued vaccination, if needed. This will set the agenda for future research to address gaps in knowledge about the determinants of vaccination behaviors going forward and the potential utility of the VaCCS as an important tool in this endeavor. We could only collect data on a USA sample as at the time of data recruitment only a small section of Australian residents were eligible to get the COVID-19 vaccine (e.g., those aged 50 years and over; Aboriginal and Torres Strait Islander peoples aged 18 years and older; those with underlying medical conditions; and those whose employment placed them at high risk of contracting or spreading COVID-19). By contrast, all USA residents aged 18 years and older were eligible. 

REVIEWER COMMENT: It is unclear to me, how this paper contributes to the literature. Is it through the creation and validation of a serious of indicators which might explain Vaccination Concerns in COVID-19 Scale? I would recommend the authors to highlight the contribution and state it (or their research question clearly).

AUTHORS’ RESPONSE: Thank you, we agree it is important to characterize the study aims as clearly as possible. We now state: “To date, there is no evidence-based measure that captures the sets of beliefs and concerns expected to be related to COVID-19 vaccination intentions and, ultimately, actual vaccination uptake. This study addresses this evidence gap by developing and validating a comprehensive measure that captures these concerns, the Vaccination Concerns in COVID-19 Scale (VaCCS). The VaCCS will provide researchers and practitioners with a novel measure to assess beliefs and concerns toward COVID-19 vaccination and assist in identifying the determinants of vaccination intention and uptake, informing the development of messaging and interventions that may promote vaccination. Importantly, our approach to validation is intended to produce a scale that is flexible to use such that researchers and practitioners can use the scale in its entirety or select specific sub-scales to use according to their needs.” 

REVIEWER COMMENT: The choice of the items for the VaCCS are backed-up by very few references, which appear quite surprising considering that the authors have done a systematic review to choose them. I would suggest them to provide a few more reference so they can bolster their methods.

AUTHORS’ RESPONSE: All the references from the rapid review process and from which scale items for the VaCCS development were extracted are provided in Supplemental Material B. We regret this was not previously made available to Reviewers, but this has now been remedied – the supplemental materials should be available as an attachment, and also online. We have specified that data files, analysis output, and supplemental materials are available online: https://osf.io/k96bn/. 

REVIEWER 2

REVIEWER COMMENT: This is a very timely and important study. The methods are well detailed and robust. I have made some suggestions for improvement in the attached pdf as comments. My main suggestion is that the methods are redundant in some places (the same thing is explained several times), and several sections could be consolidated.

AUTHORS’ RESPONSE: We thank Reviewer 2 for their positive comments and suggestions, which has helped us further develop this research. We have addressed each of their comments in the itemized list below.

REVIEWER COMMENT: No page. Need an OSF ID to access the files

AUTHORS’ RESPONSE: We regret this omission, it was an oversight because we did not make our OSF page ‘public’. However, we have now rectified this and Reviewers should have full access to our materials. 

REVIEWER COMMENT: Page 2. Specify what theory was used to guide scale development

AUTHORS’ RESPONSE: There was no specific theory that guided scale development, therefore we have removed this term from the abstract. 

REVIEWER COMMENT: Page 2. What software was used?

AUTHORS’ RESPONSE: Given journal word limits for the Abstract we have not included the software used to run analyses. Instead, we have included this detail in the Method and Data Analysis sub-section.

REVIEWER COMMENT: Page 3. Define vaccine hesitancy

AUTHORS’ RESPONSE: A good point, we have now provided a definition of vaccine hesitancy: “One contributing factor may be vaccine hesitancy, which represents a psychological state of indecision with respect to getting vaccinated [27].”

REVIEWER COMMENT: Page 3. 'populist' is not the right term here - that propagate misinformation would be more accurate

AUTHORS’ RESPONSE: We have now replaced ‘populist’ with ‘misinformation’.

REVIEWER COMMENT: Page 4. Specify the difference between vaccine hesitancy and concerns and false beliefs about COVID-19 vaccines.

AUTHORS’ RESPONSE: A good point, we now state: “Vaccine hesitancy in general, and specific concerns and false beliefs with respect to the COVID-19 vaccines, have considerable potential to stymie vaccination program effectiveness.” 

REVIEWER COMMENT: Page 4. Provide a few examples of such beliefs - both COVID-19 specific as well as "other" beliefs - not clear if you are referring to anti-vaxers in general (against all vaccines, not specifically against the covid-19 vaccine) or religious beliefs, for example.

AUTHORS’ RESPONSE: We have now provided some examples: “In addition to vaccine hesitancy in general, other beliefs (e.g., social and moral norms) [24, 25] and concerns (e.g., trust in government) [26] specific to COVID-19 may also contribute to intentions to receive, and actual uptake of, COVID-19 vaccines.” 

REVIEWER COMMENT: Page 4. I would use the term "vaccine coverage" rather than "reach"

AUTHORS’ RESPONSE: Done.

REVIEWER COMMENT: Page 4. Of

AUTHORS’ RESPONSE: Done.

REVIEWER COMMENT: Page 4. this definition should be stated earlier on

AUTHORS’ RESPONSE: Done.

REVIEWER COMMENT: the

AUTHORS’ RESPONSE: Done.

REVIEWER COMMENT: Page 6. This process is described in detail below and this repetitive here - I suggest consolidating

AUTHORS’ RESPONSE: Thank you. We have now consolidated this section to reduce repetition of information presented in other sections. 

REVIEWER COMMENT: Page 6. Was this process driven by theory? If so, which theory? 

AUTHORS’ RESPONSE: The process was not driven by a specific theory but rather by a systematic, multi-step design process using the AMEE Guide. The AMEE Guide presents a seven-step survey scale design process broadly consisting of a development phase (steps 1-6) and a validation phase (step 7). We have now made this clear in the Methods section. 

REVIEWER COMMENT: Page 6. Which specific existing scales served to inform the current scale?

AUTHORS’ RESPONSE: All the references from the rapid review process and from which scale items for the VaCCS development were extracted are provided in Supplemental Material B. We regret this was not previously made available to Reviewers, but this has now been remedied – the supplemental materials should be available as an attachment, and also online. We have specified that data files, analysis output, and supplemental materials are available online: https://osf.io/k96bn/.

REVIEWER COMMENT: Page 6. and about how COVID-19 emerged

AUTHORS’ RESPONSE: Thank you, we have now added this text. 

REVIEWER COMMENT: Page 8. significance test of differences among groups should be presented in an additional column.

AUTHORS’ RESPONSE: We have added a column in the sample characteristics table that provides overall tests of difference for the characteristics across samples and their significance values. We also report these tests of difference and the accompanying follow-up tests in the results section. 

REVIEWER COMMENT: Page 9. This seems repetitive from what was stated above in the methods / suggest consolidating.

AUTHORS’ RESPONSE: Reviewer 1 requested we expand on descriptions of the design phases, which we have done, although we have tried to keep repetition across the sections in the manuscript to a minimum. 

REVIEWER COMMENT: Page 10. Software used?

AUTHORS’ RESPONSE: Done.

REVIEWER COMMENT: Was this rubric or assessment based on some tool? Or was it developed by the researchers? Specify

AUTHORS’ RESPONSE: This measure was developed by the researchers; we now state this in the revised manuscript.

REVIEWER COMMENT: parentheses are unbalanced

AUTHORS’ RESPONSE: Done.

REVIEWER COMMENT: I find a lot of repetition in the methods - I recommend consolidating and shortening

AUTHORS’ RESPONSE: We have tried to consolidate and shorten sections in the revised manuscript while also trying to address the comments raised by both reviewers. 

REVIEWER COMMENT: Page 20. citation missing

AUTHORS’ RESPONSE: Done.

REVIEWER COMMENT: present if group differences are significant

AUTHORS’ RESPONSE: We have now extended this section to report tests of difference and the accompanying follow-up tests for the demographic characteristics in the results section.

REVIEWER COMMENT: Mention anti-vaxers in general - distrust of other vaccines

AUTHORS’ RESPONSE: We have made a brief mention that the trust subscale is likely to capture general “anti-vax” beliefs. 

REVIEWER COMMENT: Page 41. maybe free will relates more to attitudes towards vaccine mandates rather than to the vaccine itself (although these should be correlated)

AUTHORS’ RESPONSE: Thank you, we have included this suggestion in the paper: “One possibility is that free will is associated with attitudes towards vaccine mandates, such as proof of vaccine requirements for travel, work, and entry into bars and restaurants, rather than to the vaccine itself, although these beliefs would be expected to correlate.”

REVIEWER COMMENT: how is this compared to fear of COVID itself? was this assessed at all?

AUTHORS’ RESPONSE: Here we are measuring fear of getting the vaccine rather than generalized fear of COVID, although these factors would also be expected to correlate. We did not measure fear of COVID, so this is a speculative explanation. We have added a comment on this in the revised manuscript: “This finding is consistent with theory [81] and research [82, 83] demonstrating that uncertainty is likely to generate feelings of fear in health contexts, and perhaps fear of COVID itself although this needs empirical corroboration, and provides further evidence for the concurrent validity of this subscale.” 

REVIEWER COMMENT: Page 42. this should be counterbalanced by risk perceptions, fear of and susceptibility to covid-19 itself - was this considered? if not, mention in limitations

AUTHORS’ RESPONSE: A good point. As we did not measure these beliefs, we have provided a speculative comment and suggest the importance of future research to address this limitation: “These beliefs may also be associated with perceptions of risk and fear of COVID-19. However, these relationships were not tested in the current analysis, and we look to future studies to provide empirical verification”. 

REVIEWER COMMENT: Page 46. Non generalizability to other countries (including low resource settings) and non language speakers should also be noted)

AUTHORS’ RESPONSE: Thank you. We have added the following to the revised manuscript: “It is also important for research to replicate current findings in non-English speaking and under-resourced countries to ensure generalizability of the VaCCS more broadly.”

REVIEWER COMMENT: Were religious beliefs considered? If not, this should be noted as a limitation

AUTHORS’ RESPONSE: Done: “However, this does not rule out the possibility that beliefs and concerns salient to specific populations, or idiosyncratic beliefs and concerns relevant to smaller segments of the population, such as religious groups, exist that were not identified in the development stages, or failed to emerge in our factor analyses.”

---

## [Decision Letter · Decision Letter 1]

17 Feb 2022

The Vaccination Concerns in COVID-19 Scale (VaCCS): Development and Validation

PONE-D-21-37977R1

Dear Dr. Hagger,

We’re pleased to inform you that your manuscript has been judged scientifically suitable for publication and will be formally accepted for publication once it meets all outstanding technical requirements.

Kind regards,

Camelia Delcea

Academic Editor

PLOS ONE

Additional Editor Comments (optional):

Reviewers' comments:

Reviewer's Responses to Questions

**Comments to the Author**

1. If the authors have adequately addressed your comments raised in a previous round of review and you feel that this manuscript is now acceptable for publication, you may indicate that here to bypass the “Comments to the Author” section, enter your conflict of interest statement in the “Confidential to Editor” section, and submit your "Accept" recommendation.

Reviewer #2: All comments have been addressed

2. Is the manuscript technically sound, and do the data support the conclusions?

Reviewer #2: Yes

3. Has the statistical analysis been performed appropriately and rigorously? 

Reviewer #2: Yes

4. Have the authors made all data underlying the findings in their manuscript fully available?

Reviewer #2: Yes

5. Is the manuscript presented in an intelligible fashion and written in standard English?

Reviewer #2: Yes

6. Review Comments to the Author

Reviewer #2: The authors have satisfactorily addressed all outstanding comments. The manuscript is clearly written and methods explained fully. The science and procedures are sound.

7. PLOS authors have the option to publish the peer review history of their article (what does this mean?). If published, this will include your full peer review and any attached files.

Reviewer #2: **Yes: **Ariadna Capasso

---

## [Editor Report · Acceptance letter]

28 Feb 2022

PONE-D-21-37977R1 

The Vaccination Concerns in COVID-19 Scale (VaCCS): Development and Validation 

Dear Dr. Hagger:

I'm pleased to inform you that your manuscript has been deemed suitable for publication in PLOS ONE. Congratulations! Your manuscript is now with our production department. 

Kind regards, 

on behalf of

Dr. Camelia Delcea 

Academic Editor

PLOS ONE